# Characterization of an RNA binding protein interactome reveals a context-specific post-transcriptional landscape of *MYC*-amplified medulloblastoma

Michelle M. Kameda-Smith[1,2,3], Helen Zhu[4,5,6,7], En-Ching Luo[8,9,10], Yujin Suk [1,2,11], Agata Xella[12], Brian Yee [8,9,10], Chirayu Chokshi[1,2], Sansi Xing[1,2], Frederick Tan[8,9,10], Raymond G. Fox[13], Ashley A. Adile[1,2], David Bakhshinyan[1,2], Kevin Brown [14], William D. Gwynne[1,2], Minomi Subapanditha[1], Petar Miletic[1,2], Daniel Picard [15], Ian Burns[11], Jason Moffat [14], Kamil Paruch[16,17], Adam Fleming [18], Kristin Hope[1,2], John P. Provias[19], Marc Remke [15], Yu Lu[1,2], Tannishtha Reya [13,22], Chitra Venugopal[1,3], Jüri Reimand [4,5,20], Robert J. Wechsler-Reya [12,22,23] ✉, Gene W. Yeo [8,9,10,23] ✉ & Sheila K. Singh [1,2,3,21,23] ✉

Pediatric medulloblastoma (MB) is the most common solid malignant brain neoplasm, with Group 3 (G3) MB representing the most aggressive subgroup. *MYC* amplification is an independent poor prognostic factor in G3 MB, however, therapeutic targeting of the MYC pathway remains limited and alternative therapies for G3 MB are urgently needed. Here we show that the RNA-binding protein, Musashi-1 (MSI1) is an essential mediator of G3 MB in both *MYC*-overexpressing mouse models and patient-derived xenografts. *MSI1* inhibition abrogates tumor initiation and significantly prolongs survival in both models. We identify binding targets of MSI1 in normal neural and G3 MB stem cells and then cross referenced these data with unbiased large-scale screens at the transcriptomic, translatomic and proteomic levels to systematically dissect its functional role. Comparative integrative multi-omic analyses of these large datasets reveal cancer-selective MSI1-bound targets sharing multiple MYC associated pathways, providing a valuable resource for context-specific therapeutic targeting of G3 MB.

Medulloblastoma (MB) is the most common solid malignant brain neoplasm in children comprising four subgroups: Wingless (WNT), Sonic Hedgehog (SHH), Group 3 (G3), and Group 4 (G4)[1]. G3 MB is the most aggressive subgroup[2], and due to its propensity to disseminate throughout the leptomeningeal axis it is associated with a poor prognosis[3,4]. Multiple high-throughput genomic studies identified a subset of G3 MB tumors that uniquely harbors *MYC* amplification events that independently stratify risk for G3 MB patients[5,6]. Whereas targeting bromodomains[7], aurora kinases[8], and histone deacetylases[9] using small molecule therapies may inhibit MYC-associated pathways to abrogate MB progression or recurrence after upfront therapy[10], effective modalities that target MYC itself are not yet available, highlighting the need to identify alternative mediators of G3 MB that can be exploited therapeutically.

Recent advances in proteogenomics have suggested a requisite role of post-transcriptional modifications in G3 MB, including the *MYC*-amplified subtype (Group 3 *γ*)[3,11,12]. Discordance between the transcriptome and proteome in G3 MB have suggested a central role of post-transcriptional gene regulation such as alternative splicing, alternative polyadenylation, RNA turnover and export[13–17]. While genomic, epigenomic, and proteomic platforms continue to facilitate analyses of primary tumor tissue, the ability to directly link transcriptomic changes to the proteome remains a significant challenge. Overcoming these barriers is essential in studying neoplasms of the brain, which exhibits higher levels of post-transcriptional gene regulation required for neuronal development, plasticity and memory[15,16].

Musashi-1 (MSI1) is an RNA-binding protein (RBP) that maintains the multipotentiality of neural stem/progenitor cells during development, becoming rapidly downregulated in post-mitotic neurons[18,19]. Similarly, cancer stem cells with high clonogenic potential[20–22] in neural and non-neural tissues were found to highly express multiple neural stem cell determinant genes[23–25] including MSI1[26–43]. Since its discovery, multiple experiments have been designed to determine a functional role for MSI1 within the nervous system. Until recently however, biochemical protocols to reproducibly identify in vivo binding targets of an RBP with high fidelity limited any experimental conclusions. The development of irreversible RBP cross-linking and immunoprecipitation and sequencing[44] facilitated the generation of translationally relevant hypotheses of RBP function. Notably, for MSI1, this technique facilitates comparative analysis of RBP-mediated mechanisms that govern neural stem cells during normal brain development by comparison to brain cancer stem cells during G3 MB tumor initiation.

Brain cancer stem cells (also termed brain tumor initiating cells; BTICs) evade conventional adjuvant therapies[45–47] to fuel disease progression, and expression of antigens that mark BTIC are correlated with poor clinical outcomes for brain tumor patients[48]. However, therapeutic targeting of stem cell genes such as MSI1[42,49–52] that serve as normal developmental moderators[53] introduces unique therapeutic challenges in treating childhood brain cancer patients during critical periods of neural development. To facilitate a large-scale comparative study of MSI1 function and targetability in childhood brain cancer, we investigate its role as a mediator of stemness in neural stem cells by comparison with BTICs from endogenously MSI1-enriched *MYC*-amplified G3 MB. We follow MSI1 target transcripts through transcriptomic, translatomic and proteomic changes that emerge after genetic knockdown (KD) in patient-derived G3 MB cell lines to characterize the multi-stratum influence of a single RBP. Herein, we present a comparative, integrative, multi-omics approach that can delineate MSI1's master regulatory role unique to *MYC*-amplified G3 MB, providing a valuable resource for data-informed therapeutic target selection.

## Results

### MSI1 is enriched in Group 3 medulloblastoma and associated with the expression of brain cancer stem cell markers

To identify the clinical relevance of *MSI1* upregulation in the largest primary MB publicly-available database, we interrogate the transcriptomic profile of *MSI1* in patient-derived MB[6]. In contrast to Vo et al.'s observation of a survival disadvantage associated with higher levels of MSI1 protein on MB patient specimens[37], we did not find as strong an association between *MSI1* transcript levels and patient survival, highlighting concerns of exclusively transcriptomic-based interpretations of cancer biology and subsequent targeted drug selection (Supplementary Fig. 1a). To determine preclinical correlates of MSI1 protein levels we perform western blot analysis of MSI1 in G3 MB cell lines, which reveal greater MSI1 in these cells (Supplementary Fig. 1b, SU_MB002 and HD-MB03), as compared to the moderate expression of MSI1 observed in human neural stem cells (NSCs) cultured from the embryonic cerebellum (Supplementary Figure 1b,

NSC201cb), and low expression of MSI1 in a WNT subtype MB cell line (Supplementary Fig. 1b, BT853). To identify a relationship between MSI1 and other protein markers associated with stemness and BTICs in our MB cell lines, flow cytometric analysis of BMI1[54–56] and CD133[25] levels was conducted on patient-derived MB cells revealing enrichment of neural cancer stem cells markers in G3 MB (SU_MB002 and HD-MB03) (Supplementary Fig. 1f).

### MSI1 maintains stem cell properties in normal neural stem cells and G3 MB

To link our phenotypic observations in G3 MB BTICs to a functional role of MSI1 in neural cancer stem cells, we knocked down (KD) *MSI1* by RNA interference (sh*MSI1*) in NSCs. Inhibition of *MSI1* in NSCs (Supplementary Fig. 1c) lead to significant reductions in stem cell properties, including secondary neurosphere formation ($p < 0.001$) and proliferation ($p < 0.001$) capacities (Supplementary Fig. 1d, e). These results are corroborated by recent genome wide CRISPR-Cas9 gene knockout screens demonstrate a potent genetic vulnerability to MSI1 loss in NSCs[57]. While this observation contrasts existing literature[58], our results suggest that MSI1 may play an essential role in the developing human nervous system and, for this reason, it should be weighed if considered as a direct therapeutic target for treatment of childhood MB. Instead, identifying G3 MB specific, downstream protein-coding transcripts modulated by MSI1 may present more pragmatic therapeutic avenues.

### MSI1 is required for tumor propagation in PDX and murine models of G3 MB

To investigate the role of MSI1 in *MYC*-amplified G3 MB, we initially utilize a well-established G3 MB mouse model, driven by overexpression of <u>M</u>yc and dominant-negative <u>p</u>53 (hereafter called the MP model)[59]. Mouse Msi1 protein levels were greater in MP tumors than in adult mouse cerebellum and olfactory bulb, but lower than in embryonic brain (Supplementary Fig. 2a). Mouse *Msi1* transcript levels are also significantly upregulated in MP-derived tumors compared to adult mouse cerebellum and comparable to mouse NSCs ($p = 0.048$) (Supplementary Fig. 2b). Knockdown of Msi1 in MP cells by RNA interference (sh*Msi1*) (Supplementary Fig. 2c) result in abrogation of neurosphere formation (Supplementary Fig. 2d–e) and reduction in proliferation potential (Supplementary Fig. 2f). To determine whether *Msi1* was necessary for tumor propagation, we use previously described *Msi1*^flox/flox mice[60] (Supplementary Fig. 2g–h). MP tumors are generated from neonatal *Msi1*^flox/flox mouse NSCs, and tumor cells are infected with retroviruses encoding GFP (control) or Cre recombinase (for *Msi1* excision) (Supplementary Figure 2i). As shown in Supplementary Fig. 2j, Cre diminish *Msi1* mRNA levels compared to control cells ($p = 0.0015$). Msi1 deletion impair the ability of tumor cells to proliferate and form primary and secondary neurospheres (Supplementary Fig. 2k–l). Finally, Cre-infected *Msi1*^flox/flox MP cells are intracranially engrafted to determine effects of *Msi1* deletion on tumor growth (Supplementary Fig. 2i). Strikingly, whereas mice transplanted with control tumor cells develop tumors within two months, none of the mice transplanted with Cre-infected *Msi1*^flox/flox tumor cells show evidence of tumor formation (Fig. 1a, Supplementary Fig. 2m) and exhibited 100% survival over 300 days (Fig. 1b). As Cre-mediated cellular toxicity and induction of complete tumor regression has been described[61], similar studies are repeated using MP tumors from wild type (non-*Msi1*^flox/flox) mice. No significant differences in survival were seen between mice receiving Cre-infected and GFP-infected tumor cells, suggesting that the survival benefit in *Msi1*^flox/flox mice was not secondary to Cre toxicity (Supplementary Fig. 2n).

Building on observations of *Msi1* deletion in a syngeneic mouse *Myc*-amplified G3 MB model, we further determine if similar tumor suppressive effects would be achievable in human models of *MYC*-amplified G3 MB. Given the central role of MSI1 in maintaining

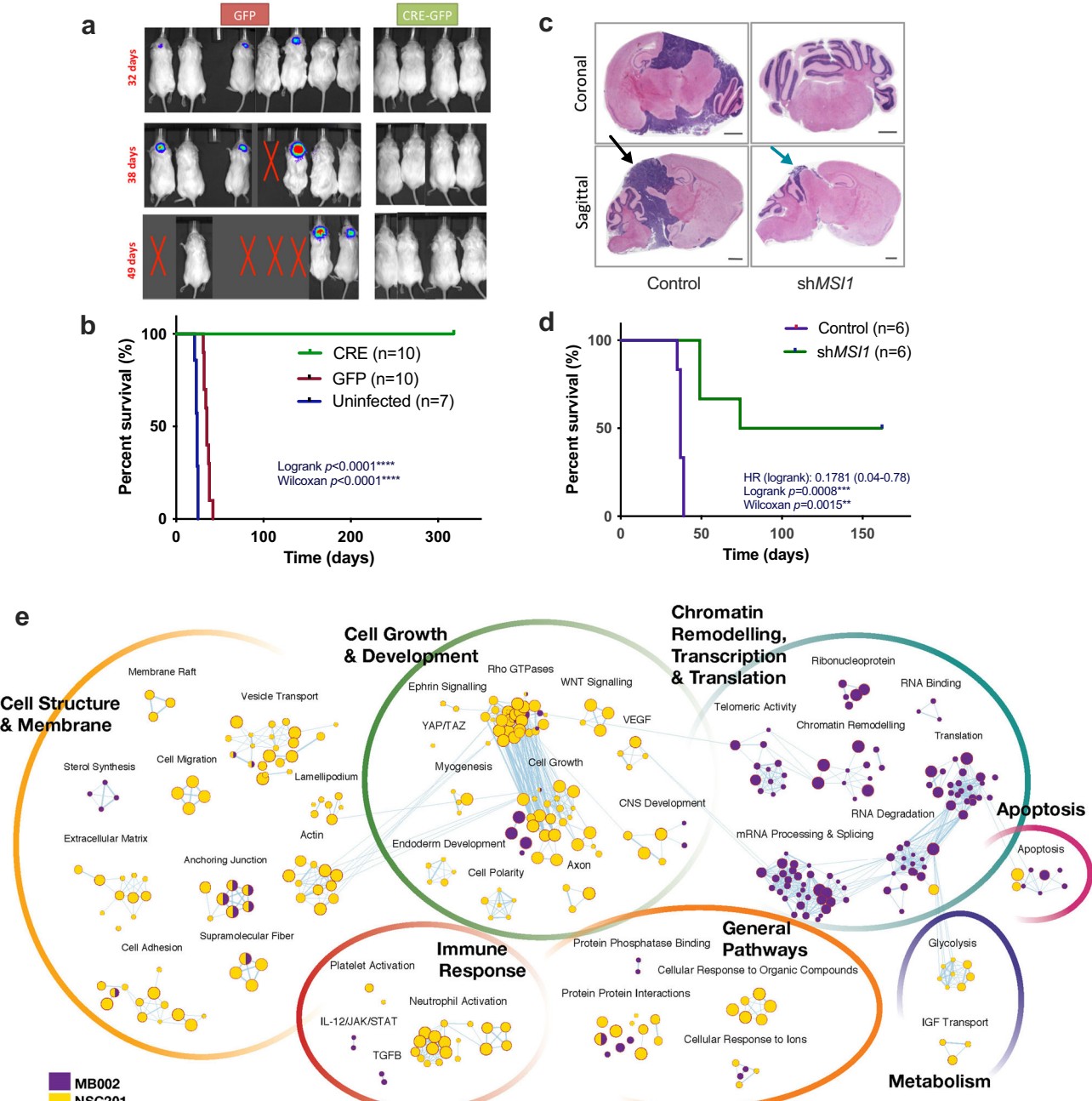

**Fig. 1 | *MSI1* is overexpressed in G3 MB. a** Tumor burden analysis of mice ortho-topically transplanted with Cre-infected Msi1^flox/flox showing no luminescence in Cre mediated *Msi1 KO* MP tumors indicating impaired tumor formation compared to GFP control mice, **b** *Kaplan-Meier* survival analysis of Cre-mediated *Msi1* KO mice ($n = 10$), control GFP mice ($n = 10$) and untransduced MP mice ($n = 7$) illustrating a clear survival benefit for the Cre-mediated *Msi1* KO cohort ($p < 0.001$), **c** 24 NOD-SCID mice were engrafted with SU_MB002 with $2.5 \times 10^4$ cells. H&E stain of control *vs* sh*MSI1* KD SU_MB002 transplanted mice in the tumor burden arm demon-strating large tumors in the control *vs* sh*MSI1* harvested brains (bars represent 100 μm), **d** *Kaplan-Meier* survival curve of control *vs* sh*MSI1* KD SU_MB002 trans-planted mice ($p < 0.0008$), **e** Pathways analysis of differentially MSI1 eCLIP bound transcripts identifying key processes that are aberrant in SU_MB002 as compared to NSC201cb (>2logFC).

stemness, and a recent SU_MB002 RNAi screen identifying MSI1 to be essential in SU_MB002[62], shRNA interference is employed to study the role of MSI1 in G3 MB in vitro. Western immunoblotting show significant MSI1 protein depletion in SU_MB002 cells trans-duced with sh*MSI1* lentivirus, despite modest *MSI1* KD at the tran-script level (Supplementary Fig. 3a). Functional assays including secondary sphere formation and proliferation show significant impairment of stem cell properties after sh*MSI1* inhibition in G3 MB BTICs (Supplementary Fig. 3b–c). Notably, despite high transcript abundance of *MSI1* in SU_MB002, further *MSI1* overexpression show

additional amplification of proliferative capacity that is suggestive of an *MSI1* gene dosage effect (Supplementary Fig. 3d). To investi-gate the effect of sh*MSI1* inhibition in vivo, immunocompromised mice are injected intracranially with control ($n = 12$) or sh*MSI1* inhibited ($n = 12$) SU_MB002 cells. Large tumors are observed in control mice, as is deduced by hematoxylin and eosin (H&E) stain-ing of xenografts, whereas sh*MSI1* G3 MB engrafted mice exhibit substantially reduced tumor burden (Fig. 1c). Deficits in tumor initiation of sh*MSI1* mice translate into a significant survival benefit as visualized by *Kaplan-Meier* survival analysis ($p < 0.0008$)

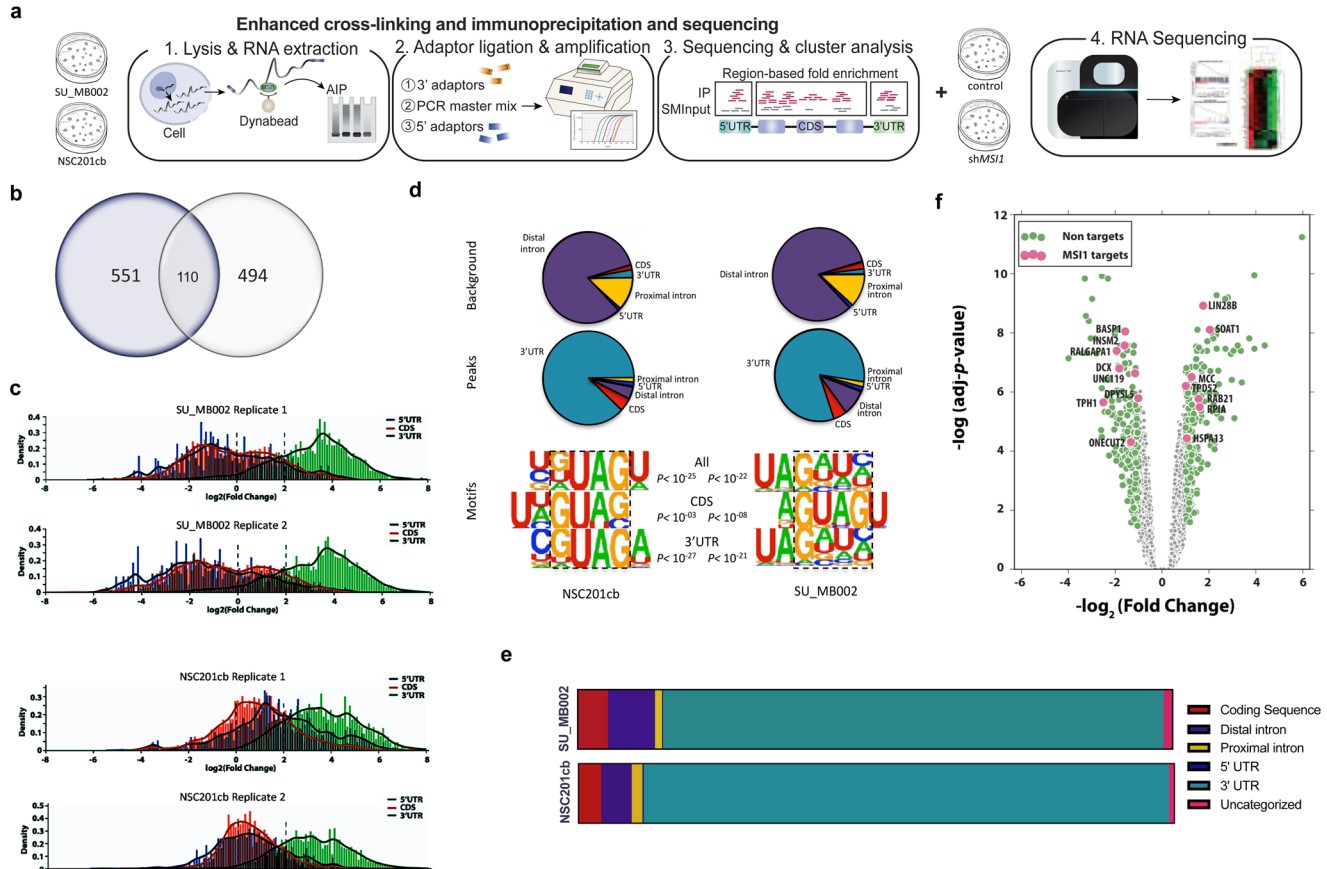

**Fig. 2 | eCLIP analysis reveals key MSI1 bound transcripts involved in processes associated with chromatin remodeling, transcription and translation.**
**a** Schematic of experimental plan to identify MSI1 binding in SU_MB002 and NSC201cb and examination of changes to steady state transcriptome with RNA-seq, **b** Venn diagram showing moderate overlap of binding genes between SU_MB002 and NSC201cb (>3logFC), **c** eCLIP peak calling identifies 3′UTR (green peaks) as highest confidence binding sites for both SU_MB002 and NSC201cb (>3logFC), d. Motif analysis identifying MSI1 consensus binding sequence as G(UAG) with highest confidence in the 3′UTR ($p < 10^{-21}$) in SU_MB002, **e** Bar graph showing proportion of binding regions in NSC201cb and SU_MB002, **f** Volcano plot presenting genes that are bound by MSI1 are modestly differentially expressed (mean logFC 0.19 ± 0.52, Range = −3.69–2.10).

(Fig. 1d), where 50% of mice in the sh*MSI1* cohort are asymptomatic at time point sacrifice and devoid of any histological evidence of tumor at the time of sacrifice. Observations noted in SU_MB002 are supported in a second metastatic G3 MB cell line (HD-MB03) depleted of *MSI1* (Supplementary Fig. 3e). Large tumors are observed in mice engrafted with HD-MB03 cells in the control ($n = 12$) as compared to the sh*MSI1* cohort ($n = 12$) (Supplementary Fig. 3f). This tumorigenicity difference lead to a significant survival benefit in sh*MSI1*-engrafted mice ($p = 0.0002$) (Supplementary Fig. 3g). Together, our data suggest MSI1 is a regulator of human NSC and is required for tumor propagation in human G3 MB. As MSI1 has been demonstrated as a vital normal neurodevelopmental gene of human NSC, inhibiting MSI1 directly would be predicted to be associated with significant toxicity. Instead, we employ a multi-platform approach to identify differentially binding targets of MSI1 in G3 MB and NSC, elucidating tumor-specific targets for future therapeutic design.

Unraveling the MSI1-mediated post-transcriptional landscape of key genes associated with neural stemness and G3 MB using multi-omic MSI1-eCLIP, RNA-seq, polysome profiling and TMT-MS.

## eCLIP analysis reveals key MSI1 bound transcripts involved in processes associated with chromatin remodeling, transcription, and translation

In order to systematically characterize the MSI1 post-transcriptional regulon specific to MYC-amplified G3 MB, we perform a differential analysis in G3 MB (SU_MB002) and embryonic cerebellar NSCs (NSC201cb) using enhanced crosslinking, immunoprecipitation and sequencing (eCLIP-seq)[44] (Fig. 2a). By using NSCs that are cultured from the matched anatomical origin of MB as a control, we hypothesize that candidate G3 MB-specific downstream targets of MSI1 can be identified. Replicates highly correlate via gene reads per kilobase of transcript per mapped reads (RPKM) with 1271 and 1382 binding sites mapping to 551 and 494 unique gene symbols in G3 MB and NSC respectively (Fig. 2b; Supplementary Tables 1 and 2). These neoplastic and normal cells of cerebellar origin shared 110 MSI1 binding transcripts (Fig. 2b; Supplementary Table 3). Included in shared genes bound by MSI1 are *CTNNB1* and *eIF4* subunits (i.e., *A2* and *G2*), suggesting MSI1's role in moderating genes involved in essential processes such as transcription and translation initiation (Supplementary Fig. 4a). A statistically significant fraction of genes in these cell lines overlap (OR 13.615, Fisher's exact test, $p < 2.2 \times 10^{-16}$), indicating shared patterns of vital post-transcriptional regulatory processes. Furthermore, a similar pattern of gene region specificity from reproducible MSI1 peaks is observed in G3 MB and NSCs as bound sequences were primarily in the 3′UTR region, followed by the coding sequence (CDS) and 5′UTR (Fig. 2c–e; Supplementary Table 4). Motif analysis verify the previously reported G(UAG)U trinucleotide is enriched around MSI1 eCLIP sites in both G3 MB and NSC[40,63] (Fig. 2d). Differential analysis of genic regions bound by MSI1 in G3 MB and NSC identify unique G3 MB binding transcripts (e.g., *MYC, OTX2, BRD3, DDX3X*) that may represent potential downstream targets (Supplementary Fig. 4b), in addition to

shared targets (e.g., *CTNNB1*, *TCF12*, *EIF4A2*) that may represent remnants of normal neural stem cells (Supplementary Fig. 4a; Supplementary Table 3). Subsequent pathway analysis identify 551 and 494 MSI1-bound genes in G3 MB and NSC, respectively (Supplementary Table 5). Among the MSI1-bound transcripts in uniquely binding to G3 MB, we identify an enrichment of genes associated with known G3 MB pathways (e.g., JAK-STAT, TGF-β)[64–66], as well as genes associated with chromatin remodeling, transcription and translation pathways in G3 MB as compared to NSCs (Fig. 1e; Supplementary Fig. 4c; Supplementary Tables 6 and 7) indicating a departure from normal MSI1 function in G3 MB. Overall, the differential binding of transcripts by MSI1 between G3 MB and NSC, its presumed cell of origin, suggests an aberrant MSI1-associated post-transcriptional modification of essential genes in G3 MB.

## RNA-seq of shMSI1 G3 MB identifies downregulation of sets of genes within known G3 MB-associated pathways

To identify whether inhibition of *MSI1* directly alters the steady state mRNA landscape and offer insight into MSI1's role in mRNA processing and stability, we perform RNA-sequencing (RNA-seq) of sh*MSI1*-inhibited and control G3 MB cells. Despite testing significance relative to threshold (TREAT)[67], between control and experimental expression of microarray data, modest inhibition of *MSI1* result in 10,686 transcripts with detectable expression differences of which 1580 were significantly differentially expressed (adjusted $p < 0.05$). Gene set enrichment analysis (GSEA) show downregulation of known G3 MB associated signaling pathways, including TGF-β and GABA receptor activation (Supplementary Fig. 4d–e)[1,5,6,64,65,68,69]. Notably, a large number of eCLIP-bound genes are modestly differentially expressed after sh*MSI1* inhibition, suggestive of an extensive cascading effect (i.e., downstream effects of MSI1 binding transcripts) of sh*MSI1* inhibition. Ninety-one MSI1 eCLIP-bound genes are however differentially expressed after sh*MSI1* inhibition suggestive of MSI1 in regulating transcript stability or degradation (Fig. 2f). The transcriptomic changes that are observed after the inhibition of a single RBP gene may be explained in part by binding of transcripts of 119 other annotate RNA-binding proteins (e.g., *MATR3, HNRNPA1/PA3, NONO, PABPC1*)[70,71] (Supplementary Table 9) and 7 lncRNAs (e.g., *MALAT1, SNHG6, RP11, OIP5*) transcripts to MSI1, and their subsequent effect throughout the transcriptome (Supplementary Table 10). Differential analysis at the level of the transcriptome offer insights into the vast sequelae of MSI1 inhibition. However, in order to achieve a more holistic understanding of MSI1's function, it became clear that further interrogation of RBP translational regulation was essential.

## Polysome profiling and sequencing of shMSI1 G3 MB reveals translational downregulation of key cancer associated genes

As 91 eCLIP-bound genes are significantly differentially expressed after sh*MSI1* inhibition in the RNA-seq data, we characterize the translatome to determine whether MSI1 mediates access of these transcripts to ribosomes. We explore the changes occurring in the polysome fraction after sh*MSI1* inhibition in G3 MB to determine if there is a significant change in the fraction of mRNA associated with polyribosomes (i.e., MSI1-mediated accessibility of mRNA to ribosomes) (Fig. 3a). The polysome profile of the sh*MSI1*-inhibited samples demonstrate a relative global increase in polysome-associated mRNA as compared to the control sample (Fig. 3b). Overall, polysome-seq identify 11,385 transcripts that are differentially associated with the polysome fraction with input-normalized RPKM values (Supplementary Table 11). A significant fraction of the mRNA that are differentially polysome-associated in sh*MSI1* as compared to control MB cells are annotated as cancer associated genes (Fig. 3c). Multiple cancer-associated genes are significantly downregulated in the translatome upon *shMSI1* inhibition, including *MAP3K13*, transcriptional enhancers of *MYC* and genes essential in early

development, *NPM1* and *KMT2A*, respectively. To further investigate those genes whose expression is increased in the sh*MSI1* samples (i.e., mRNA stalled in translation or exported, localized and translated together as part of the same mRNA regulon)[72–74], we initially quantify the nascent peptide formation in the sh*MSI1* and control samples. We employ a FACS-based method of the surface sensing of translation (SuNSET) assay[75] and observe an overall global increase in nascent polypeptide production after *MSI1* inhibition (Fig. 3d, Supplementary Fig. 5a–d). Overall, the downstream effect of *MSI1* KD is an increase in translation of transcripts associated with DNA repair and transcriptional regulation by TP53 (Supplementary Fig. 6a–b), implying that after sh*MSI1* inhibition, differentiation, and apoptotic cascade initiation results in increasing the polyribosome association with transcripts governing cell survival.

## Proteomic analysis of shMSI1 G3 MB identifies upregulation of genes associated with neuronal differentiation and down-regulation of cancer aggressiveness

Finally, to understand which proteins are translationally promoted, shotgun quantitative proteomic analysis is performed on control and sh*MSI1* inhibited G3 MB cells (Fig. 3e). MaxQUANT analysis of our isobaric compound tagged sample peptides identified through liquid chromatography and mass spectrometry (LC-MS) show a right shift in abundance of protein upon MSI1 KD compared to control, suggesting an overall increase in protein abundance (Fig. 3f, Supplementary Fig. 7a–b, Supplementary Table 12). Further analysis identify 561 proteins with significant differences in abundance levels, the majority of which (i.e., 76.57%) showed fold change >1 after sh*MSI1* inhibition (Fig. 3g). Protein set enrichment analysis (PSEA)[76] reveal downregulation of proteins associated with cancer aggressiveness and an upregulation of proteins associated with neuronal differentiation and apoptosis after sh*MSI1* inhibition (Fig. 3h). These findings corroborate the polysome profiling and sequencing results and support MSI1's known role as a stem cell determinant in both neural and cancer stem cells.

## Integrative multi-omics analysis identifies HIPK1 as a potential downstream cancer-selective target in G3 MB

Given the striking phenotypic consequences of genetic MSI1 depletion, we reason that accompanying changes in gene expression may be leveraged to identify drugs that could be repurposed to treat G3 MB. To this end, connectivity map (CMAP) analysis of the RNA-seq data comparing control *vs* sh*MSI1* G3 MB cells (SU_MB002), identified a number of repurposable drugs, including Lomustine (i.e., currently in use for MB chemotherapy) that parallel the transcriptomic profile after sh*MSI1* inhibition in G3 MB (Supplementary Fig. 8a). To test whether these top compounds would be able to abrogate G3 MB function while sparing NSCs, we perform multiple $IC_{50}$ assays. As hypothesized, the therapeutic window was narrow, with few concentrations adequately targeting G3 MB whilst sparing NSC function (Supplementary Fig. 8b–e) suggesting a significant risk of on-target, off-tumor effects predicted from our in vitro results after inhibition of MSI1 in NSCs. Therefore, we turned to the multiple data sets generated from multiple biochemical techniques for further candidate gene selection. By employing integrative pathway enrichment analysis, we examine the convergence of differentially expressed genes observed within the transcriptome, translatome and proteome belonging to key cellular processes. Given differences associated with contrasting experimental platforms used in multi-omic studies, in lieu of applying a single statistical cut-off across platforms, a robust rank aggregation (RRA) for gene list integration[77] was employed to identify candidate genes for G3 MB MSI1-bound downstream target selection. Comparison of the RRA list of genes with those identified using single platforms highlight overlapping and distinct pathways (Fig. 4a). Pathways associated with eCLIP-

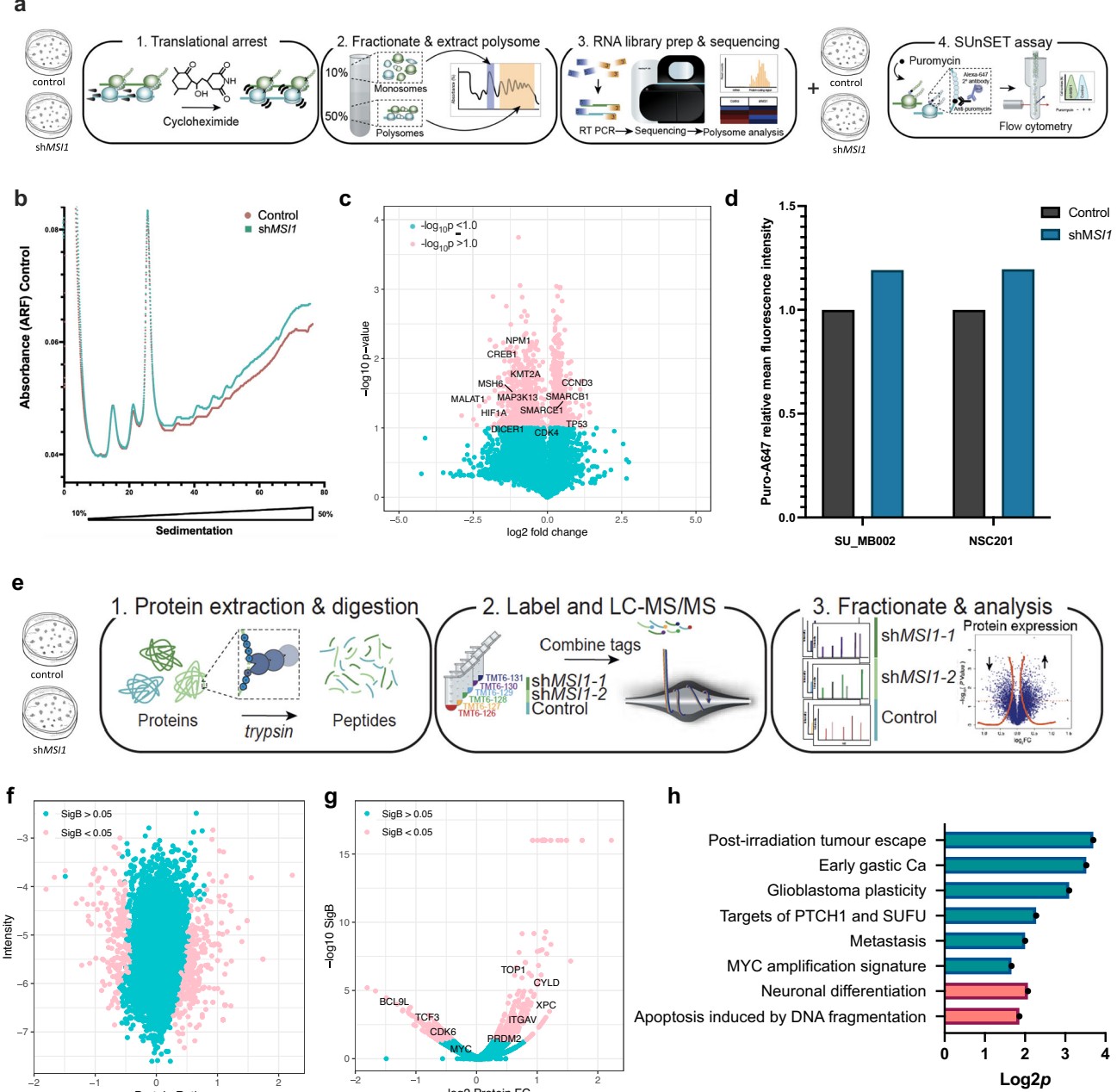

**Fig. 3 | Polysome profiling and sequencing and SuNSET assay suggests MSI1 is aberrantly impeding the association of transcripts with ribosomes with subsequent whole shot gun proteomic analysis identifying downregulation of proteins associated with cancer aggressiveness. a** Schematic of experimental plan to investigate the ribosome associated fraction of mRNA and validation of global translational process with SuNSET assay, **b** Polysome profile of control *vs* sh*MSI1* SU_MB002 illustrating an increased mRNA polysome associated fraction in sh*MSI1* inhibited samples, **c** Polysome-seq identifies key cancer associated genes that are down regulated in the polysome fractions after sh*MSI1* inhibition in SU_MB002, **d** SuNSET assay after normalization shows a trend towards increased nascent polypeptide production in sh*MSI1* inhibited samples in SU_MB002 and NSC201, **e** Schematic of experimental plan to quantitatively investigate the proteomic landscape after sh*MSI1* inhibition, **f** MaxQUANT and PERSEUS analysis of TMT-MS data showing 350 significantly differentially abundant proteins in the sh*MSI1* samples compared to the control (Significance B < 0.05), **g** Volcano plot with annotation of key genes that are up and down regulated at the proteomic level with ~50% inhibition of the MSI1 protein, **h** PSEA analysis identifying downregulation of proteins associated with cancer aggressiveness in green, and upregulation of genes associated with neuronal differentiation and DNA fragmentation in red (log2*p* > 1).

binding in SU_MB002 were previously described in Fig. 1. At the transcriptomic level, we observe a disruption in vital functions of the neural cancer stem cell such as neurogenesis and synaptic signaling. Major pathways perturbed by *MSI1* inhibition at the translatomic level are associated with post-transcriptional processes in addition to with cell survival (i.e., autophagy, glycolysis). Proteomic analyses identify proteins associated with unfolded protein response and

wound healing, indicating an overall attempt by the cell to regain homeostasis following a decrease in MSI1 function.

This comprehensive data-driven approach using RRA analysis identify 224 genes that are significantly altered by *MSI1* perturbation (RRA Score < 0.01) (Fig. 4a right panel, Supplementary Table 13). Candidate genes are compared to genes identified by the Cancer Gene Census (CGC)[78] and previous MB literature[1,5,6,64,69,79–84]. Overlap of

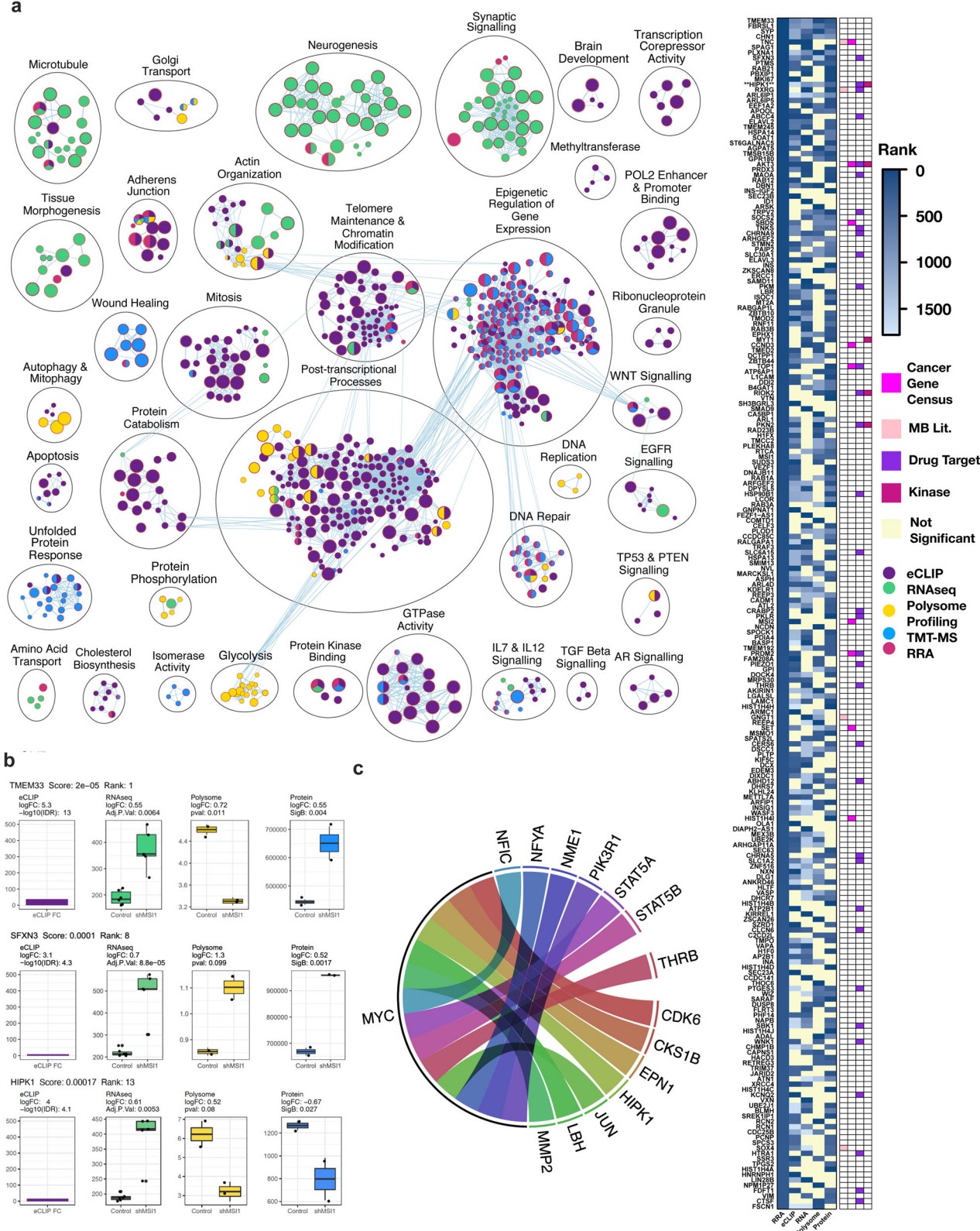

**Fig. 4 | Robust ranked analysis (RRA) identifies TMEM33, SFNX3 and HIPK1 gene set directly perturbed by MSI1 in G3 MB. a** Pathways analysis illustrating the main pathways perturbed by the binding of MSI1 to its target (Purple:eCLIP, Green:RNA-seq, Yellow:polysome-seq, Blue: proteomics (TMT-MS), Pink: RRA). The associated heat map shows the RRA rank and gene association (Yellow denotes non-significant association) with a cancer annotated gene (fuchsia) or MB associated gene (salmon), **b** TMEM33, SFNX3 and HIPK1 changes after sh*MSI1* inhibition in G3 MB), **c** Circos plot showing significantly perturbed genes after sh*MSI1* inhibition and their regulation by MYC (Reactome Functional Interaction).

cancer-associated genes with our data set highlighted nine CGC and four G3 MB associated genes that intersected with the 224 genes identified by our RRA analysis. This finding is suggestive of MSI1's twin purpose in dysregulating known cancer-associated genes (i.e., *TNC*, *AKT3*, *CCND3*, *MSI2*) in addition to genes associated with normal functioning of the cell (i.e., *TMEM33*, *SYN*, *MK167*).

RRA analysis further identify a set of MSI1-bound genes in G3 MB, including *TMEM33*, *SFNX* and *HIPK1*, as cancer-selective downstream targets for therapeutic drug targeting (i.e., not bound in NSC) (Fig. 4b left panel, Supplementary Fig. 9a). In *TMEM33* and *SFNX3*, there is congruence between the positive changes at the transcriptomics and proteomics data; However, the changes in *HIPK1* expression is positively regulated by *MSI1* inhibition but with a negative effect at the protein level. The RRA results are further supported by transcriptomic analyses of primary tumor specimens[6,85], which reveal a positive correlation between *MSI1* and *HIPK1* expression in MB (Fig. 4b, Supplementary Fig. 9b) highlighting the importance of investigating not only the genomic and transcriptomic landscape, but the post-transcriptional changes when studying the functional role of an RNA binding protein. Interestingly, interrogation of the Reactome functional interaction network[86] show MYC-mediated regulation of the expression of multiple top RRA genes, including *HIPK1* (Fig. 4c, Supplementary Table 14). Moreover, the multiple annotated pathways of HIPK1 converged with 114 biological pathways associated with the proto-oncogene MYC (Supplementary Table 15) suggestive of an ancillary MYC suppression effect via targeting HIPK1. Given that HIPK1 is not ubiquitously expressed throughout the body as compared to MSI1 (ProteomeDB.org), targeting HIPK1 in MB may provide a context-specific therapeutic strategy.

To further validate our multi-omics platform to identify a clinically distinct G3 MB context specific target for therapeutic discovery, we interrogate the effect of targeting HIPK1 in MSI1 high expressing G3 MB cells (i.e., SU_MB002). *HIPK1* CRISPR knock out (KO) constructs are transduced into G3 MB and NSC cell lines. Validating our multi-omic platform and target identification analyses, *HIPK1* pooled KO result in a striking reduction of cancer stem cell functions in SU_MB002 (Supplementary Fig. 10a–c) and HD-MB03 (Supplementary Fig. 10d–f) (G3 MB cell lines), whose stem cell properties are rescued with *HIPK1* overexpression after *MSI1* knockdown (Supplementary Fig. 10g), while sparing human NSC (Supplementary Fig. 12a–d). Clonal *HIPK1* KO in SU_MB002 and HD-MB03 (Supplementary Fig. 11a–e) in vivo conferred a striking survival benefit and tumor burden (Supplementary Fig. 12e–g). These corroborate Toledo et al., genome wide CRISPR KO data showing HIPK1 to have a non-essential Bayes Factor score in multiple NSC cell lines providing further validation of the potential neurodevelopmental safety profile our target (Supplementary Fig. 12h). These observations are in accordance with our discovery of G3 MB associated MSI1-mediated regulation of *HIPK1* transcripts (Fig. 4a right panel, Supplementary Fig. 9a) and provides a robust rationale for further pre-clinical investigation of HIPK1 for therapeutic drug discovery in G3 MB.

## Discussion

Following the generation of extensive datasets from multiple platforms, the challenge to analyze and extract clinically meaningful results for targeted drug discovery has been substantial. Computational integration of genomic, transcriptomic, and proteomic datasets has been evolving over the last decade with significant analytical challenges overcome by insights provided by mapping distinct molecular signals to known pathways and biological processes[87–89]. These advances have enabled the unbiased systematic investigation of post-transcriptional regulators to better delineate their role in cancer and identify potential therapeutic targets. Applied to descriptive genomics-based studies of both primary tumor tissue and cell lines, advancements in the field of bioinformatics have uncovered important insights into the probable cells of origin, driver genes, and heterogeneity within histologically similar tumor entities. However, to date, targeted therapies identified using these approaches alone have fallen short of an effective cure for these children. Cancer is characterized by changes in cell symmetry and self-renewal capacity and therefore it comes as no surprise that genes regulating these processes in normal neural development (i.e., MSI1) may be implicated in developmental brain tumors such as MB.

Patients harboring MYC-amplified tumors show consistently poor prognosis and survival[5,6,11]. Challenges of achieving safe, complete surgical resection of MB, adverse effects of exposing pediatric patients to whole brain and spinal radiation, and incomplete response to conventional chemotherapies stresses the need for a deeper and multi-level understanding of this disease. By leveraging insight from recent proteomic studies in pediatric MB[3,11,12], we performed an in-depth multi-omic analysis of G3 MB BTICs to uncover the master regulatory role of the RBP MSI1. We aimed to identify the key transcripts MSI1 modulates to regulate multiple biological nodes at a post-transcriptional level, ultimately affecting self-renewal and tumorigenicity of G3 MB stem cells. The striking tumor suppressive effect of MSI1 inhibition in our syngeneic and PDX models of MB provides evidence for the necessity of MSI1 in G3 MB stem cell self-renewal and propagation. In keeping with previous studies that introduced a crucial role for MSI1 in MB[28,38,39], we show compelling in vitro and in vivo confirmation that consolidates the breadth of literature on MSI1 as an essential post-transcriptional gene modifier in G3 MB. Globally, our multi-platform and integrative analysis suggest a departure from the normal MSI1 modulated gene set (i.e., in NSC) that is post-transcriptionally modified in G3 MB. Considering the combined impact of 800–1000 known human RBPs[70,71] has recently been estimated to account for as much as 30% of protein expression variation[90–92], we show that post-transcriptional modulation of transcripts by MSI1 in part explain previously described discrepancies between the transcriptome and proteome in G3 MB[3,11]. For example, MSI1's binding of hundreds of transcripts in G3 MB, including the most frequently mutated gene in G3 MB (*SMARCA4*) and characteristic G3 MB genes such as *MYC* and *OTX2*[1,5,6,64,69,79–81,84], imply that MSI1 may stabilize mutated transcripts and enable greater activity of their oncogenic protein products (Supplementary Fig. 4b) possibly through the upregulation of alternative splicing[93]. Using our multi-omics platforms, we continue to build and develop methods to filter through extensive datasets towards the identification of effective and safe tumor-specific driver genes.

The search for novel cancer driver genes for targeted drug discovery remains a significant challenge in pediatric tumors such as MB, particularly because many genes associated with tumorigenesis are implicated in normal developmental processes. In previous transcriptomics-based studies of cancer, Notch signaling has been associated with MB[69,94,95], with a number of studies suggesting MSI1 binding to the Notch pathway inhibitor and determinant of cell fate and neurodevelopment, *NUMB*[96–100] as an overarching inhibitory signal of NOTCH pathway gene products[101]. Despite experimental evidence of Msi1 binding to *m-Numb* and the conclusion that Msi1 protein is likely involved in the translational repression of m-Numb protein expression, we did not find evidence of this interaction in our neural stem cells (G3 MB and NSC). As supported by Uren et al.[40] Sakakibara et al.[58] and Katz et al.[93], in glioblastoma, mouse neural progenitor and mouse E12.5 neural stem cells from transgenic mice with a Dox-inducible Msi1 allele respectively, we found that while *NUMB* was expressed in our system, it was not eCLIP-bound by endogenous MSI1 in SU_MB002 nor NSC201cb. In fact, downregulation of the Notch pathway upon MSI1 inhibition did not follow the MSI1-*Numb* axis but led to downregulation of key genes associated with the Notch pathway and the polysome fraction (Supplementary Table 11). To give greater credence to our eCLIP observations, we compared our list to

those of other studies performing MSI CLIP studies in cancer. After confirming the consensus MSI1 binding motif GUAGU[102] was negligibly bound to stop codons (Supplementary Table 16), a comparative analysis revealed 99.9% of SU_MB002 and NSC201cb MSI1-eCLIP hits were shared with glioblastoma and hematopoietic stem and progenitor cells (HSPCs) (Supplementary Fig. 13a–b)[40,103], further validating our eCLIP data set. Altogether, these observations reemphasize the importance of direct studies of endogenously expressed gene and/or protein product, rather than transgenic approaches, to study its activity.

Our stringent and integrative analysis across multiple experimental platforms identified HIPK1 as the top MSI1-downstream G3 MB targeting (i.e., not MSI1 bound in NSC) candidate gene for potential therapeutic drug discovery (Fig. 4a, Supplementary Fig. 9a). Functionally, HIPK1 has been observed to phosphorylate the cAMP response element binding (CREB) protein[104], whose microdeletion is associated with Rubinstein-Taybi syndrome, a known G3 MB associated disease[105,106] indicating clinical relevance of HIPK1 targeting in G3 MB. Interestingly, in our platform, *HIPK2* was MSI1-bound in both G3 MB and NSC stem cells whereas differential binding of *HIPK1* was identified only in G3 MB. Previous experiments dissecting the role of HIPK1 and its isoform HIPK2 showed the loss of HIPK2 reduces cellular responses to TGF- β during neuronal development in mouse models, while mice lacking HIPK1 showed no detectable defects in neuronal development[107–109]. These findings corroborate our pre-clinical data suggesting HIPK1 represents a robust and potentially neurodevelopmentally safer downstream MSI1-binding target for drug development in G3 MB. Furthermore, our comparative approach was taken with the hypothesis that differential MSI1 bound target transcripts result in divergent cellular modifications with potential functional consequences in G3 MB *vs* NSC. Considering high MSI1 expression in both fetal and adult neural stem cells, and with future goals of therapeutic safety in mind, we confirmed MSI1's essential role in normal neural stem cells. In contrast to Sakakibara et al.'s observations that *Msi1* deletion alone did not decrease self-renewal ability of mouse NSCs[58], in line with Chavali et al.'s description of consanguineous twin study with clinical features suggestive of autosomal primary microcephaly (i.e., significant reduction in cerebral cortex size, but a structurally normal brain) with whole exome sequencing uncovered potentially deleterious homozygous mutations in *MSI1*. Furthermore, we found that sh*MSI1* inhibition in human NSCs led to significant reduction in stem cell properties of proliferation and secondary sphere formation (Supplementary Fig. 1c–d), with sparing of stem cell properties in a MSI1's downstream target, HIPK1 in NSC (Supplementary Fig. 12a–d), with a significant survival benefit in mice injected with multiple G3 MB cell lines with CRISPR deletions of *HIPK1* gene expression (Supplementary Fig. 12e–g). This is in keeping with in silico data identifying a high Bayes Factor score for MSI1 and low score for HIPK1 in a CRISPR screen of NSCs (Supplementary Fig. 12h). These findings suggest while MSI1 proves to be a desirable target for neoplastic lesions within the brain, it also plays a central role in maintaining functional stem cell properties of human NSCs and should therefore be avoided with a therapeutic strategy designed to target a MSI1 downstream G3 MB unique substrate in the developing brain.

In summary, our panoramic and unbiased integrative analysis reveals a data-driven downstream MSI1 post-transcriptional target, HIPK1 for therapeutic drug discovery. Furthermore, we have shown our multi-omic approach has the powerful potential to facilitate future drug discovery beyond traditional limitations of targeting ubiquitously expressed and essential proteins such as MSI1. Our current work highlights how an integrative comparative multi-omics approach focused on deep characterization of a single RBP provides levels of information not accessed by transcriptomic characterization alone. By identifying high confidence binding targets of an oncogene followed by unbiased multi-platform analysis, the ability to advance context-specific biological discovery and therapeutic targets become feasible.

## Methods

### Experimental model and subject details

**Cell culture of G3 MB cell lines.** Primary human pediatric MBs, SU_MB002 and HD-MB03 were kind gifts from Dr. Yoon-Jae Cho (Harvard, MS) and Dr. Till Milde (Heidelberg) respectively (Table 1). SU_MB002 is primary G3 MB cell line developed from a sample acquired at autopsy from a 4-year-old boy who was metastatic at presentation. Given his poor post-operative clinical status, parents had turned down radiotherapy and he received post-operative adjuvant cyclophosphamide. He initially responded to treatment only to experience disseminated disease within 3 months and palliated. HD-MB03 is G3MB line established from a fresh tissue section obtained at therapeutic intervention from a 3-year-old boy who was metastatic at presentation as previously described[110]. The primary human MB, BT853 is a Wnt MB cell line established from fresh tissue section at surgical resection from a 5-year-old female. BT853 was established after informed consent from the family and as approved by the Hamilton Health Sciences/McMaster Health Sciences Research Ethics Board. The sample was dissociated in PBS containing 0.2 Wünsch unit/mL of Liberase Blendzyme 3 (Roche), and incubated at 37 °C in a shaker for 15 min. The dissociated tissue was filtered through a 70-µm cell strainer and collected by centrifugation (1200 rpm, 3 min). Tumor cells were resuspended in a serum-free BTIC enrichment media, and replated on ultra-low attachment plates (Corning). BTIC enrichment media was composed of NeuroCult complete media (StemCell Technologies, 10 ng/mL bFGF, 20 ng/mL EGF, 2 µg/mL heparin). Expansion media was used prior to experimentation and BTIC enrichment. SU_MB002 and was expanded using the same BTIC enrichment media. HDMB-03 was expanded with the BTIC enrichment media supplemented with 10% fetal bovine serum (FBS). BT853 was expanded with Dulbecco's Modified Eagle's Medium high glucose (Life Technologies #11965-118) supplemented with 10% FBS. All samples were cultured in BTIC enrichment media for at least 48 hours prior to experimentation.

**Cell culture of MP tumors.** The development and culturing of MP tumors have been previously described[59]. MP tumors were in vivo expanded in vivo in 6–8 week old NOD-SCID cerebellum. Cells from the mouse cerebellum were harvested after the animal reached endpoint (i.e., clinical: any head swelling, animal is quiet, circling, 10% weight loss from pre-injection). On the same day as tumor processing, After purification, neural stem cells and tumor cells were maintained in vitro at $1 \times 10^6$/mL cells in Neurocult basal media supplemented with 10% Proliferation supplement (Stem cell technologies), Penicillin/Streptomycin 1% (Thermo Fisher Scientific), basic Fibroblast growth factors (bFGF, 25 ng/mL, PeproTech) and Epidermal growth factor (EGF, 25 ng/mL, PeproTech). Cells were counted automatically with TC10 automated cell counter (Biorad). 4-hydroxytamoxifen (4OHT, 1 M in DMSO) was used as stock solution for 4OHT experiments.

**Animal husbandry and in vivo experiments.** All in vivo experiments were performed in accordance to the McMaster University Animal Research Ethics Board (AREB) approved protocols national guidelines and regulations, and with the approval of the animal care and use committees at SBP and at the University of California San Diego (UCSD). UCSD Protocol S12123 and Sanford Burnham Prebys Protocol 14-027. In vivo experiments were performed in accordance to the McMaster University Animal Research Ethics Board (AREB) approved protocols national guidelines and regulations, and with the approval of the animal care and use committees at SBP and at the University of California San Diego (UCSD).

For the MP *Msi1*^fl/fl experiments, WT (C57BL/6 J) pups for tumors generation were obtained from the Sanford Burnham Prebys (SBP) Medical Discovery Institute Animal Facility. NSG (**N**OD-**S**CID IL2R-**g**amma null) mice were purchased from Jackson Labs were used as hosts for orthotopic tumor transplantation. Mice were maintained in

**Table 1 | Resource table of reagents, samples, software and algorithms employed**

| Reagent or resource | Source | Identifier |
|---|---|---|
| **Antibodies** | | |
| MSI1 | Abcam | ab52865, RRID: AB881168 |
| HIPK1 | Abcam | ab90103, RRID: AB_2041622 |
| BMI1 | R&D | MAB33342 |
| Nestin | Millipore | AB5922 |
| CD133/2-APC, human clone REA820 | Miltenyi | 130-112-196 |
| β-tubulin | Abcam | ab6046 |
| GAPDH | Abcam | ab8245, RRID:AB_2107448 |
| Puromycin | Millipore | MABE343-AF647, RRID: AB_2736876 |
| **Biological samples** | | |
| SU_MB002 | Bandopadhayay et al.[7] | RRID: CVCL_VU79 |
| Neural stem cells | This study | |
| HD-MB03 | Milde et al.[110] | RRID: CVCL_S506 |
| MP tumors | Pei et al.[59] | |
| **Critical commercial assays** | | |
| Super Script II cDNA synthesis Kit | Invitrogen | Cat: 18064-014 |
| Quant –iT dsDNA BR Assay Kit | Invitrogen | Cat: Q32850 |
| NEBNext Ultra II Directional RNA Library Prep Kit for Illumina | NEB | Cat: E7760S |
| **CRISPR-Cas9 vectors** | | |
| lentiCRISPRv2 | Addgene | Cat: 52961 |
| **Deposited data** | | |
| MS data | This study | PRIDE: PXD012432 (ProteomeXchange PRIDE database) |
| Raw and processed RNA-seq data | This study | GEO: GSE126337 |
| Raw and processed Polysome-seq data | This study | GEO: GSE134597 |
| Raw and processed eCLIP data | This study | GEO: GSE126263 |
| **Software and algorithms** | | |
| STAR | Dobin et al., 2013 | https://github.com/alexdobin/START/releases |
| Bowtie 2 | Landmead and Salzberg, 2012 | http://bowtie-bio-sourcefourge.net/bowtie2/index.shtml |
| R | | https://r-project.org/ |
| GSEA | Subramanian et al., 2005 | http://software.broadinstitute.org/gsea/index.jsp |
| GO | Chen et al., 2013 | http://geneontology.org/docs/go-enrichment-analysis/ |
| EnrichR | Kuleshov et al., 2016 | |
| Samtools merge v1.6 | Li et al., 2009 | http://samtools.sourceforge.net/ |
| g:Profiler | Reimand et al., 2007 | https://biit.cs.ut.ee/gprofiler/ |
| ActivePathways | Paczkowska et al., 2018 | https://github.com/reimandlab/ActivePathways/ |
| Cytoscape (v.3.6.0) | Shannon et al., 2003 | http://www.cytoscape.org |
| Boutroslab.plotting.general (v.5.9/2) | P'ng et al., 2019 | https://labs.oicr.on.ca/boutros-lab/software/bpg |
| Robust ranked aggregation (RRA) | Kolde et al., 2012 | https://cran.r-project.org/web/packages/RobustRankAggreg/index.html |

the animal facilities at the Sanford Consortium for Regenerative Medicine (SCRM). Primary MP tumors were generated as described in Pei et al.[59]. CD133[+] neural stem cells from postnatal day 4-6 mouse cerebella were purified by percoll fractionation and FACS sorting with phycoerythrin (PE)-conjugated rat anti-CD133 antibody (clone 13A4, eBioscience, 1:200) in FACS Buffer (Dulbecco's PBS + 5% fetal bovine serum, Invitrogen/Life Technologies); the top 5% of the CD133[+] population was selected. Cells were infected with retroviruses encoding *Myc* and *DNp53* (see Retroviruses below) overnight and transplanted into the cerebellum of 6- to 8-week-old NSG mice $(8 \times 10^4 - 1.5 \times 10^5$ cells/mouse) using a stereotaxic apparatus (see Tumor and Stem Cell Transplantation below). The number of live cells was determined using Trypan Blue (Thermo Fisher) and automated counting using a TC10 automated cell counter (Biorad). Cells were resuspended in Neurocult media (Stem cell technology) to achieve the desired number of cells per injection (5 µL). Stem cells and tumors were transplanted into the cerebellum of 6–8 week old mice using a

stereotaxic frame. Mice were anesthetized using isoflurane gas (5% induction) in a designed chamber; the head was secured using one nose- and two ear-holds. While in the stereotaxic apparatus, mice were kept anesthetized by isoflurane flow at 2.5%. The skull overlying the cerebellum was exposed and a craniostomy was performed -3 mm caudal of the lambdoid suture and 2 mm off the sagittal axis. A 5 µL Hamilton syringe loaded with 5 µL of a previously prepared solution of cells was angled at 30 degrees in the sagittal line and inserted 3 mm deep into the hole. Cells were slowly injected, pausing after half of the solution was injected. After the procedure, the incision was sutured and reinforced using surgical glue. The mice were then given 0.5 mL of analgesic subcutaneously. Animals were monitored with in vivo bioluminescence imaging and euthanized when they showed signs of MB or reached experimental endpoint (i.e., clinical: any head swelling, animal is quiet, circling, 10% weight loss from pre-injection).

For the human sh*MSI1* experiments intracerebellar injections were all performed by the first author via free hand technique using

anatomical landmarks (i.e., 5 mm caudal to lambda and to the right 5 mm laterally) and injected at a depth of 3 mm using a 10 μL Hamilton syringe for SU_MB002 and HDMB-03. The number of live cells was determined by using Trypan Blue (Thermo Fisher) exclusion and resuspended in 5uL of BTIC enrichment media. NOD-SCID mice were anesthetized using isoflurane gas (5% induction, 2.5% maintenance) and the cells were injected into right cerebellar hemisphere. Tumor-initiating capacities of SU_MB002 and HD-MB03 comparing control and sh*MSI1* knockdown constructs were performed by injecting $5.0 \times 10^5$ and $1.0 \times 10^3$ cells respectively. Mice were assessed for histological differences of tumor burden ($n = 6$ in each arm) and survival (additional $n = 6$ in each arm).

**Retro/Lentiviral production.** Retroviruses were used to generate tumors and induce deletion of flanked-by-loxP (floxed) genes. MSCV-Myc-IRES-CD2 and MSCV-DNp53-IRES-Luciferase viruses were used to create primary tumors. MSCV-CRE-IRES-GFP was used for Cre-mediated excision of floxed sequences, and MSCV-IRES-GFP was used as a control. Viruses were generated by calcium chloride mediated transfection of 293T cells. We plated 293T cells in 10 cm plates: 10 mL/plate at $1.8$-$2 \times 10^5$ cells/mL in Dulbecco's media (DMEM with L-Glutamate 4.5 g/L glucose with sodium pyruvate supplemented with 10% fetal bovine serum and 1% Penicillin/Streptomycin, Thermo Fisher). After 24 h we changed the media, adding only 9 mL per plate. 1–2 h after media change, cells were transfected with 1 mL per plate of a solution composed of: 0.5 mL HEPES 7.3 1 M, 62 μl of 2 M Calcium Chloride, 10 μg Gag-Pol, 4 μg VSVG, 6 μg retroviral construct. Media was changed and discarded after overnight incubation. Subsequently, we performed 3 collections, one every 24 h. The supernatant collected was filtered with a 0.4μm filter and ultracentrifuged at $63,000 \times g$ for 2 h at 4 °C using Optima L-80 XP Ultracentrifuge. Viral pellets were resuspended in 0.2–1.4 mL of Neurocult media (Stem cell technologies). Optimal viral concentration was tested for each batch in 293T cells by FACS or Steady glow assay (Promega) for the luciferase-bearing virus.

Human *MSI1* shRNA were developed in the Dr. Jason Moffat laboratory (Lentiviral TRC RNAi library, University of Toronto). Human oeMSI1-CFP (Genocopia) and human oeHIPK1 (OmicsLink™) was purchased commercially. Replication-incompetent lentiviruses were produced by co-transfection of the expression and packaging vectors pMD2G and psPAX2 in HEK293FT cells. Viral supernatants were harvested 72 h after transfection, filtered through a 0.45 μm cellulose acetate filter, and ultracentrifuged at $42,000 \times g$ at 4 °C for 2 h. The viral pellet was resuspended into 200 μL of DMEM media and stored in −80 °C. Cells were analyzed for all in vitro and in vivo studies 96 hours after transduction and puromycin selection. Mouse *Msi1* shRNA were developed in the Dr. Tannishitha Reya laboratory. Viruses for cell transduction were produced with the same protocol as the Human *MSI1* shRNA.

**RT-qPCR.** For mouse samples, total RNA was isolated from cells using the Zymo RNA miniprep kit (Zymo Manufacturing company) according to the manufacturer's instructions. Up to 1 μg of template RNA, in

accordance with the sample availability, was reverse transcribed to cDNA using iSCRIPT (Biorad) following protocol. qPCR was carried out using SYBRGreen (Biorad) on a BioRad CFX384 thermal cycler. Primer sequences provided in Table 2.

For human samples, total cellular RNA was isolated using the Norgen Total RNA isolation kit and quantified using a NanoDrop Spectrophotometer ND-1000. Complementary DNA was synthesized from 0.5–1.0 μg RNA using iScript cDNA Super Mix (Quanta Biosciences) and a C1000 Thermo Cycler (Bio-Rad) with the following cycle parameters: 4 min at 25 °C, 30 min at 42 °C, 5 min at 85 °C and hold at 4 °C. RT-qPCR was performed using Perfecta SybrGreen (Quanta Biosciences) and a CFX96 instrument (Bio-Rad). Products were quantified by ΔΔCt analysis normalizing to GAPDH/β-actin expression, and data were presented as mean and standard error of the mean. Statistical analysis was performed using graph Pad software applying one-way ANOVA. Primers are listed in the table below.

**Western immunoblotting.** For mouse samples, 1–10 μg of protein per well was run on a 10% SDS gel. After transfer and blocking with 5% milk in TBST, immunoblotting was performed using the following primary antibodies: MSI1 (rat, 1:200, eBioscience #14H1), alpha-tubulin (rabbit, 1:1000, Cell Signaling #11H10). IRDye® 800CW goat anti-rat IgG (1:5000), IRDye® 680RD goat anti-rabbit (1:10,000) and IRDye® 800CW goat anti-mouse IgG (1:5000) were employed as the secondary antibodies (LICOR). All antibodies were diluted in 5% milk in TBST.

For human samples, denatured total cell protein (10 μg) was separated using 10% Bis-Tris gel electrophoresis and transferred to nitrocellulose membranes. Western blots were probed with the following antibodies: β-tubulin (rabbit; 1:50,000; Abcam #ab6046), GAPDH (mouse; 1:2000; Abcam #ab8245), MSI1 (rabbit; 1:2000; Abcam #ab52865) HIPK1 (1:500; Abcam #ab90103) (Table 1), Horseradish peroxidase conjugated with goat anti-rabbit IgG (1:20,000) or Licor anti-mouse (800 channel)/rabbit (700 channel) IgG were employed as the secondary antibodies (Bio-Rad). The bands were visualized with ImageStudio (Licor) or Chemidoc™ MP Imaging Systems (Bio-Rad) using the ImageLab version 15.2.1 software.

**Functional stem cell assays**

**MP Neurosphere assay.** MSI1^{fl/fl} MP tumor cells were purified, dissociated, and incubated for 24 h with MSCV-CRE-IRES-GFP retrovirus (CRE group) or GFP control (GFP group). 24 h after infection cells were gently dissociated mechanically and sorted for GFP by flow cytometry. Uninfected cells were used to gate the for GFP positive selection. Sorted cells were counted and then plated in enriched Neurocult media. 24-well plates were used, 0.5 mL media per well, 1000–2000 cells per well to ensure clonal density as described before. Growth factors (EGF and FGF) were replenished after 3 days. Neurosphere aggregates were counted, measured, and imaged 5–7 days after plating with the help of fluorescent light to identify GFP⁺ cells.

**Secondary colony formation assay.** SU_MB002, HD-MB03 tumor aggregates were mechanically dissociated with a 1000 μL pipette tip whereas MP tumorspheres and human NSC neurospheres were

**Table 2 | RT-PCR primers**

| Gene | Forward Sequence | Reverse Sequence |
|---|---|---|
| *Msi1* | 5' - TGT CTG TGA ACA CCA CGG TG – 3' | 5' - CGT GAC AAA TCC AAA CCC TCT – 3' |
| *MSI1* | 5' - CAC CAA TGG GTA CCA CTG AA – 3' | 5' - ACT CGT GGT CCT CAG TCA GC – 3' |
| *MYC* | 5' - AAT GAA AAG GCC CCC AAG GTA GTT ATC C – 3' | 5' - GTC GTT TCC GCA ACA AGT CCT CTT C – 3' |
| *GAPDH* | 5' - TGA ACC ACC AAC TGC TTA GC – 3' | 5' - GGC ATG GAC TGT GGT CAT GAG – 3' |
| *β-actin* | 5' - CCG AGC GTG GCT ACA GCT TC – 3' | 5' - ACC TGG CCG TCA GGC AGC TC – 3' |
| *β-ACTIN* | 5' - TAT CCC TGT ACG CCT CT – 3' | 5' - AGG TCT TTG CGG ATG T – 3' |
| *HIPK1 gRNA* | 5' - TCC CTT TTC ACT CCA AAT AGT TGT – 3' | 5' - ATG GTG AGC ACT ACC CTC CA – 3' |

enzymatically dissociated using TrypLE™ (Thermo Fisher). Cells were live sorted at a density of 100–500 cells per well in 200 µL of BTIC enrichment media.

**Cell proliferation assay.** Single cells were plated in 96 wells, at a density of 100–2000 cells/200 µL per well in quadruplicates for each sample and incubated for five days. 20 µL of a fluorescent cell metabolism indicator, Presto Blue (Life Technologies), was added to each well 4 h prior to the readout. Fluorescence was measured with a FLUOstar Omega Fluorescence 556 microplate reader (BMG Labtech) at an excitation and emission wavelength of 540 and 570 nm respectively. Resultant readings were analyzed using the Omega software.

**FACs sorting.** sh*Msi1*-GFP transduced MP cells were dissociated and suspended in PBS + 0.5 M EDTA + 1%FBS. Samples were sorted using MoFlo XDP cell sorter (Beckman Coulter). Dead cells were excluded using the viability dye near IR Live/Dead™ fixable staining kit (Life technologies #L10119). Compensation was performed using mouse IgG CompBeads™ (BD #552843). External and internal staining were performed as previously described[111]. Analysis probed for CD133 with an anti-CD133 human clone REA820 (Miltenyi, #130-112-196) and BMI1 with an anti-BMI REA438 c (Miltenyi, #130-106-736) stem cell markers. GFP expression was defined as positive or negative based on the analysis of regions established by the isotype control. Cells were sorted into 96-well plates containing 180 µL of mouse Neurocult complete media (Stemcell Technologies). Small aliquots from each sort were analyzed to determine the purity of the sorted populations. Cells lone equilibrated a 37 °C overnight prior to experimentation. Cells were stained for 20 min with Near-IR live stain and mCherry-GFP-live sorted for in vitro experimentation. SU_MB002 cells were stained for 5 min with 7AAD live stain (Beckman Coulter #A07704) for in vitro experimentation. Gating strategy available on Supplementary Fig. 14.

**In vivo imaging.** Luciferase-based bioluminescence imaging was performed on MP tumors as previously described[59]. Briefly, mice were given intraperitoneal injections of 150 ng/g D-Luciferin (Caliper Life Sciences, #12279) and anesthetized with 2.5% isoflurane followed by 0.2 mL of Luciferin (30 mg/mL, Caliper Life Sciences) intraperitoneal injection; mice were imaged 8 min after injection. 8 min after injections, animals were imaged using the Xenogen Spectrum (IVIS®−200) imaging system. Living image software was used to analyze results. In accordance with UCSD guidelines, mice showing signs of tumor growth (domed head, hunched posture, reduced mobility, 20% weight loss) were euthanized.

The mouse MRI imaging was performed on an automated 7Tesla wide-bore nuclear magnetic resonance (NMR) system (Bruker WB300). The protocol allows for tumor visualization without the typical necessity of injectable gadolinium contrast agents. The mouse is anesthetized in an induction chamber at 5% isoflurane in pure O$_2$ and subsequent positioning in the imaging bed with continuous anesthesia with 1.5–2.5% isoflurane delivered with pure O$_2$ via nose cone. With a combination of stretches, rotations and translations the images are warped into a common alignment, which allows a direct spatial comparison between an animal and a set of health controls. Alternatively, an animal at multiple time points were imaged allowing visual detection of tumors <0.5 mm diameter confirmed by histopathology.

**Immunohistochemistry.** Unstained slides mounted with formalin fixed paraffin embedded tissue were de-paraffinized in xylene, blocked in 3% hydrogen peroxide, and antigen retrieval or unmasking procedure applied for MSI1, BMI1, and NESTIN staining were performed as previously described at the University Health Network Medical Laboratory Technologies[112]. Anti-rabbit-MSI1 primary antibody (1:500, Millipore, #AB5977), anti-rabbit-Nestin (1:15,000, Millipore, #AB5922), and anti-mouse-BMI1 (1:500, R&D, #MAB33342) and incubated at room temperature for 1 h. For the BMI1 IHC, MACH 4 reagents were then applied as directed (Intermedico, #BC-M4U534L) and developed using DAB (DAKO, #K3468). And for MSI1 and Nestin, ImmPress® reagent were applied as per kit instruction. For all slides, Mayer's Hematoxylin was used to lightly counterstain and the slides were dehydrated. The slides were then mounted with MM 24 Leica mounting medium (Leica, #3801120).

**eCLIP-seq library preparation.** The eCLIP protocol was performed on SU_MB002 and NSC201cb as previously described[44]. 20 × 10$^6$ cells were washed in ice-cold PBS and UV cross-linked at 600 mJ cm$^{-2}$ on ice in a Stratalinker 2400. Cells were pelleted, lysed in lysis buffer and bound to Msi1 antibody (rabbit; 1:2000; Abcam #ab52865) bound magnetic Dynabeads™ M-280 sheep anti-rabbit beads (LifeTech, #11203D). RBP-RNA complexes were then captured on beads overnight in 4 °C. The samples were then dephosphorylated and the 5′- and 3′-phosphate groups from RNA using FastAP (LifeTech, #EF0652) and T4 PNK (NEB, #M0314L). The samples were then washed and 3′ linker ligated with barcoded RNA adaptors. The samples were then electrophoresed on a 1.5 mm 4–12% Bis-Tris gel and transferred onto a nitrocellulose membrane at 30 V at 4 °C overnight. The diagnostic membrane was developed to ensure adequate pulldown of MSI1 associated RNA and the lanes cut and RNA purified using acid phenol:chloroform:isoamyl alcohol (125:24:1, v/v; pH 4.5; ThermoFisher; #AM9720). The resultant RNA was cleaned and concentrated in Zymo columns (Zymo; #R1016). The RNA was then reverse transcribed using the AR17 primer: 5′ACACGACGCTCTTCCGA3′ and AffinityScript RT (Agilent, #600107) master mix. The resultant cDNA was cleaned with ExoSAPit treatment and the RNA removed. The cDNA was then 5′ linker ligated with rand3Tr3 adapter using RNA Ligase (NEB, #M0437M). After Silane bead clean of the linker-ligation, the cDNA was quantified using qPCR with subsequent PCR amplification using 2× Q5 PCR master mix (NEB, #M0492L). The DNA was then electrophoresed in a 3% low melting agarose gel and the library extracted using Qiagen MinElute gel extraction kit without heating and resuspended in nuclease-free water. The library was prepared and sent for paired-end 75 bp Illumina sequencing at McMaster University. Of the best biological replicate, 8,117,399 and 17,778,371 reads from the SU_MB002 and NSC201cb IP libraries respectively passed quality filtering, of which 57.8% and 27.2% usable reads mapped uniquely to the human genome (hg19). eCLIP data reproducibility was verified through correlation between gene RPKM and statistically significant overlaps in clusters and genes within replicates.

**RNA-sequencing (RNA-seq) library preparation.** Total RNA was extracted from each sample and purified using the Norgen Total RNA isolation kit and quantified using a NanoDrop Spectrophotometer ND-100 and quality control was assessed with bioanalyzer total RNA Nano kit. One µg of mRNA was fragmented to an average length of 200 bp by incubation for 5 min at 94 °C with 5X fragmentation buffer (Illumina, RS-100-0801). Efficiency of the fragmentation was defined on Bioanalyzer RNA Pico Chip. The fragmented mRNA was randomly primed and reversed transcribed using Super Script II cDNA synthesis kit (Invitrogen, 18064-014). After second-strand synthesis, the cDNA underwent end-repair and ligation reactions according to the Illumina mRNA-Seq Sample Prep Kit protocol. The cDNA library was size-fractioned on a 2% TBE agarose gel. Material in the 350–400 bp range was excised and purified (Zymo Research, D4001). Half of the eluted cDNA library was used as a template for amplification according to mRNA-Seq Sample Prep Kit protocol. The PCR product was purified using PureLink PCR micro purification kit (Invitrogen, Q32850). The library was then used to build clusters on the Illumina flow cell and analysis was done using Illumina Hiseq 2000 platform (Illumina, San Diego, CA, USA) at McMaster University to a target depth of 6 M reads per sample.

**Polysome-sequencing (Polysome-seq) library preparation.** $20 \times 10^6$ cells incubated for 10 min with 0.1 mg of cycloheximide (Sigma, #01810)/mL of BTIC media and incubated for 5 min at 37 °C and 5% $CO_2$. Cells were then washed and harvested in ice cold PBS and flash frozen in liquid nitrogen and stored until ready to lyse pellets. Polysome buffer (20 mM Tris HCl pH7.4, 150 mM NaCl, 5 mM MgCl2, 1 mM DTT) and lysis buffer (Polysome Buffer with 1% Triton-X + Protease Inhibitors + RNase inhibitors + cycloheximide (100 μg/mL) were prepared fresh and kept on ice. Pellets were lysed in 400 μL of Lysis Buffer per $10 \times 10^6$ cells. Lysate is then incubated on ice for 30 min then centrifuged at 15,000 rpm at 4 °C for 2 min to remove debris. The supernatant is collected and loaded onto a 10–50% sucrose gradient in Polysome Buffer with freshly added 1X Halt™ protease inhibitor (Thermo, #78444), RNAse inhibitors and 100 μg/mL of cycloheximide. 100–200 μL of cell lysate is loaded to the top of each sample column and ultracentrifuged using the Beckman SW-41 (SW 41 Ti Swinging-Bucket Rotor.) at $151,000 \times g$ at 4 °C for 3 h. Fractionation was completed using the Biocomp Model 108 Gradient Master. The total RNA is then extracted from the remaining lysate using Trizol LS (Invitrogen, #10296028). The polysome fractions are collected and pooled for polyribosome associated RNA extraction using Trizol LS. Extracted polysome associated RNA is resuspended in 50 μL of sterile ultrapure water.

One microgram of total RNA and an equivalent volume of RNA from the polysome fraction were used for library preparation as per Illumina standard RNA-seq protocol. An equal amount of Spike ins RNA Variant Controls (SIRVs, Lexogen SIRV-Set3 and ERCC) were added to each sample prior to library preparation to normalized for sequencing bias and to determine the threshold for measurable statistics. The library was then used to build clusters on the Illumina flow cell and analysis was done using Illumina Hiseq 4000 platform (Illumina, San Diego, CA, USA) at the UCSD Institute for Genomic Medicine to a target depth of 30 M reads per sample.

**Tandem Mass Tag Mass Spectrometry (TMT-MS).** Approximately 100 μg total protein was extracted from each of the six samples (replicates of SU_MB002 control -scramble cells, sh*MSI1-1* and sh*MSI-2* cells) using 8 M urea and 100 mM ammonium bicarbonate. The protein samples were reduced, alkylated, and digested by trypsin (Promega) overnight at 37 °C. The resulting peptides were desalted with 10 mg SOLA C18 Plates (Thermo Scientific), dried, labeled with 6-plex Tandem Mass Tag reagents (Thermo Scientific) before being pooled together. Sixty microgram of the pooled sample was separated into 38 fractions by reverse phase liquid chromatography (PRLC) at pH = 10 using the Thermo Acclaim PA2 C18 column (300 μm × 5 cm bed volume) packed with POROS 10R2 10 μm resin (Applied Biosystems), followed by a home-made analytical column (50 μm × 50 cm bed volume) packed with Reprosil-Pur 120 C18-AQ 5 μm particles (Dr. Maisch). LC-MS experiments were performed on a Thermo Fisher UltiMATE™ 3000 RSLCNano UPLC system that ran a 3 h gradient at 70 nL/min, coupled to a Thermo QExactive HF quadrupole-Orbitrap mass spectrometer. A parent ion scan was performed using a resolving power of 120,000 and then up to the 20 most intense peaks were selected for MS/MS (minimum ion count of 1000 for activation), using higher energy collision induced dissociation (HCD) fragmentation. Dynamic exclusion was activated such that MS/MS of the same m/z (within a range of 10ppm; exclusion list size = 500) detected twice within 5 s were excluded from analysis for 30 s.

**CRISPR-Cas9 individual HIPK1 gRNA cloning and validation.** Individual gRNA sequences were selected from the Toronto KnockOut (TKO) CRISPR Library Version 3 and cloned into the pLCV2 backbone[113] as described previously[114]. CloneAmp HiFi PCR Premix (Clontech, #639298) was used for PCR amplification. All constructs were verified using Sanger sequencing. Guide sequences targeting HIPK1 include #1

CAG ACC TTA AGC CTC CAC AG and #2 GCT GCA AGG AAA CAC CCT GC. Gene knock out (KO) validated in transduced and flow cytometric sorted pooled cells were validated using western blot prior to in vitro experimentation and clonally expanded cells using targeted sequencing prior to in vivo experimentation. Editing efficiency was evaluated by PCR amplification and Sanger sequencing of gRNA editing sites, followed by Tracking of Indels by DEcomposition (TIDE) analysis of HIPK1 knockout clones to reference *AAVS1* knockout clones[115].

## Quantification and statistical analysis

**Kaplan–Meier survival plots.** Kaplan–Meier plots were generated for the reposited Cavalli et al. dataset (using the author's annotations for Msi1$^{high}$ and Msi1$^{low}$) in GlioVis (http://gliovis.bioinfo.cnio.es/), PDX and MP data with Prism9 (GraphPad™). Log rank Hazard ratio was and both the Both Log rank (Mantel-Cox) and the Gehan-Breslow-Wilcoxan test (i.e., gives more weight to deaths at early time points). *p*-values of 0.05 or lower were considered statistically significant for all experiments.

**CMAP data processing and analysis.** CMAP analysis was used to predict the effects of currently available compounds on the expression of genes found to be eCLIP bound and differentially expressed between the control and Msi1 knockdown cells ($p < 0.0001$, Supplementary Fig. 1b)[116]. Included in this analysis were compounds that were highly associated with the ability to either potentiate or inhibit the expression of the genes in the aforementioned triprotocol analysis. CMAP analysis was conducted using Bioconductor package PharmacoGx (v3.8) using Genome Wide Correlation analysis using a seed of 314 and 100,000 permutation[117]. Genes were weighted by −log10 FDR corrected *p*-values and absolute log2 fold change. Compounds which were in agreement between the proteomic and mRNA dataset were selected. Compounds from the library were initially filtered according to their connectivity score (connectivity score > 0) and significance ($p < 0.01$). Compounds were then selected for in vitro screening in collaboration with the Broad Institute Drug Repurposing Hub[118].

**eCLIP data processing.** eCLIP reads were processed and QC was performed according to the ENCODE data processing protocol for eCLIP reads as previously described[44]. First, reads were demultiplexed according to their inline barcodes (MB002_Msi1: A01, B06; NSC201cb_Msi1: X2A, X2B) using a custom script, which also modifies each read name to include the read's unique molecular identifier (UMI) (demux.py). Next, reads were trimmed using cutadapt (v1.14) and filtered of any read mapped to RepBase (v18.05) sequences using STAR (v2.4.0j). Surviving reads were then mapped, again with STAR, to hg19 assembly to obtain genome alignments. PCR duplicate removal was then performed with a custom script based on UMI sequences placed inside each read name (barcodecollapsepe.py). De-duplicated mapped BAM files from each barcode were then combined (samtools merge v1.6), forming a single BAM file for each single IP and size-matched INPUT dataset. Read2 for each IP merged BAM file were used to call enriched peak clusters with Clipper (v1.2.1). These clusters were then normalized against size-matched INPUT reads and neighboring/overlapping clusters merged. Regions passing a −log10(p) significance of at least 3 and a log2(fold change) cutoff of 3 were deemed as significantly Msi1-bound for each replicate. The demultiplex script can be found at: https://github.com/YeoLab/eclipdemux. The pipeline definitions and barcode collapse script can be found at: https://github.com/YeoLab/eclip.

To obtain reproducible regions between two replicates, we used the modified IDR pipeline as previously described[119]. Using the outputs from the processing pipeline, input normalized peaks were ranked according to information content ($_{pi}*\log2(_{pi}/q_i)$). These ranked peaks were passed to IDR (v2.0.2) to determine regions of reproducibility. Full definitions for each tool and workflow can be found at: https://github.com/YeoLab/merge_peaks.

**eCLIP motif analysis.** Motif analysis was performed using HOMER (v4.9.1) wrapped inside a custom script (analyze_motifs.py found here: https://github.com/YeoLab/clip_analysis_legacy). The methodology was described by Lovci et al.[120]; briefly peaks were assigned to their corresponding regions of binding (CDS, 3′UTR, 5′UTR, proximal and distal intron +/− 500 bp of an exon), then compared against a randomized background (random assignments of peak coordinates across each corresponding region).

**eCLIP region-based fold-enrichment analyses.** Region-based fold-enrichment was calculated as previously described[44]. Briefly, mapped reads were counted along all transcripts in Gencode v19 ('comprehensive'). Reads were assigned to all transcripts annotated in Gencode v19. For reads overlapping>1 annotated region, each read was assigned to a single region with the following descending priority order: CDS, 5′ UTR, 3′UTR. For each gene, reads were summed up across each region to calculate final region counts. A minimum of 10 observed reads were required for a gene to be considered in region-based fold-enrichment analyses. The MSI1 eCLIP data in SU_MB002 and NSC201cb has been deposited in the NCBI Gene Expression Omnibus (GEO, http://www.ncbi.nlm.nih.gov/geo). The accession number of the eCLIP data reported in this paper is GEO: GSE126263.

**RNA-sequencing data processing and analysis.** An average of 5.5 M reads from each sample passed quality filtering. Filtered reads were mapped to the human genome (hg19) using the STAR short-read aligner (v2.4.2a) with the following command: STAR --genomeDir /path/to/GRCh37 --readFilesIn <file1.fastq.gz > <file2.fastq.gz > --readFilesCommand zcat --runThreadN 8 --outSAMstrandField intronMotif --outSAMtype BAM SortedByCoordinate --quantMode GeneCounts --sjdbGTFfile /path/to/gtf. The gencode.v19.annotation.gtf from the GENCODE database and the primary assembly of GRCh37 was used. Approximately 92% of the filtered reads mapped uniquely, and the read counts from each sample were merged into a single matrix using R. The raw and processed data has been deposited to the GEO database (accession: GSE126337).

The merged read count matrix was used to compute differential expression using the Bioconductor package limma (v3.38.3) as follows. First, transcripts were filtered using filterByExpr(min.count=10, min.total.count=15) (edgeR, v3.25.3) and normalized using calcNormFactors(method = "TMM"). A counts per million matrix was created from the normalized count matrix. Differential gene expression was conducted using the lmFit function and ranked using treat. Significant genes were identified using an FDR < 0.05 and absolute log2(foldchange)>1.

**Polysome-sequencing processing and analysis.** An average of 27 M reads from each polysome-sequencing sample were trimmed using *cutadapt* (v1.4.0) of adaptor sequences and mapped to repetitive elements (RepBase v18.04) using the STAR (v2.4.0i). The filtered reads which did not map to repetitive elements were then mapped to the human genome (hg19). Using GENCODE (v19) gene annotations and featureCounts (v1.5.0) to create read count matrices. Approximately 90% of the filtered reads mapped uniquely. The transcript RPKMs of input and polysome fractions were calculated from the read count matrices. Only genes with mean of reads≥10 and mean of RPKM ≥1 were considered. Polysome association was calculated by RPKM ratio of transcript levels in polysomes over input. The raw and processed data reported in this paper is deposited in the GEO database (accession: GSE134597).

**SUnSET assay.** The assay was performed on SU_MB002 cells were transduced with scrambled control- and shMSI1-GFP previously described[75].10[6] cells were pulsed with 10 μg/mL of puromycin for 10 min. The cells were resuspended and filtered in PBS-EDTA with 100 μg/mL cycloheximide. Cells were stained with the viability dye near IR Live/Dead™ fixable staining kit (Life technologies, cat. L10119) prior to fixing and permeabilizing with Fixation/Permeabilization Solution (BD Biosciences, #554717). Newly synthesized puromycin-tagged proteins were detected with a monoclonal primary anti-puromycin antibody (1:1000; Millipore; MABE343) and an Alexa 647-donkey anti-mouse secondary antibody (1:2000; Thermo; #A-31571) of the GFP positive cells via fluorescent activated cell sorting (FACS) of live cells.

**Proteomic data processing and analysis.** LC-MS data generated was analyzed against a UniProt human protein database (42,173 entries) for protein identification and quantification by MaxQuant software (v1.6.5) From 2,379,345 MS/MS spectra acquired in all 38 fractions, 136,833 unique peptide groups (with Peptide FDR < 0.01) and 8547 proteins (Protein FDR < 0.01) were identified and quantified[121]. The Significant B values were calculated using the PERSEUS (v1.6.5) software. Significance B value preset with an FDR < 0.01 was used to identify proteins that are significantly differentially abundant and used for downstream integrative analysis. All raw data have been deposited in the ProteomeXchange Consortium via Proteomics Identification (PRIDE)[122]. The accession number of the proteomics data reported in this paper is PRIDE: PXD012432.

**Gene ontology (GO) and gene set enrichment analysis (GSEA).** GSEA[123] was applied using a combination of MSigDB C2 curated gene sets (v6.2), C5 Gene Ontology gene sets (v6.2), and C6 oncogenic signatures (v6.2)[124]. GSEA was run using the fgsea Bioconductor package (v1.2.1). In addition, enrichment analysis was performed on sets of significant genes/proteins using the EnrichR database[125].

**Protein set enrichment analysis (PSEA).** PSEA-QUANT[76], a protein enrichment analysis algorithm was used for our label-based mass-spectrometry-based quantitative proteomics to identify protein sets from Gene Ontology and Molecular Signatures databases that are statistically enriched with abundant proteins. Abundance ratios were used, with 10,000 samplings for statistical significance assessment, annotated using the Molecular Signature Database with the assumption that protein abundance dependence in the dataset (coefficient of variation=0.5). Literature bias assuming protein annotation bias was applied.

**Triprotocol data integration.** Significant genes were selected from each of the four large-scale datasets as follows. Genes bound by MSI1 were filtered from the eCLIP binding data using an −log10IDR < 3 and log2FC (sh*MSI1*/control)>3 (8-fold). Genes differentially expressed upon *MSI1* knock-down were selected as having an adjusted *p*-value < 0.05. Significantly differentially expressed polysome associated mRNA was selected using a *p*-value < 0.1. Finally, significant proteins were selected as having a SigB < 0.05.

**Pathway analysis of eCLIP-seq data.** Pathway analysis for the comparison between eCLIP datasets in SU_MB002 and NSC201 cell lines was conducted using g:Profiler[126]. Genes were ranked by decreasing fold change. Gene sets from Reactome (v64, released 2018-10-02) and Gene Ontology databases (version Ensembl v93/ Ensembl Genomes v40, released 2018-08-03) were included. Gene sets were limited to between 5 and 500 genes and pathways were filtered for a statistical threshold of *p* < 0.05.

**Integration of datasets using Robust Rank Aggregation (RRA).** Data integration of eCLIP, mRNA, polysome-seq and protein datasets was conducted using Robust Rank Aggregation (RRA), a probabilistic

approach to aggregating rank-based lists, using the number of protein coding genes as the number of ranked elements and otherwise, using default parameters[77]. Genes were ranked by statistical significance. A threshold rho score of <0.1 was used to curate the final list. For eCLIP, the most significant peak was chosen for each gene. The significance threshold was relaxed to include sites up to $-\log_{10}IDR > 1$ and $\log_2FC > 1$. For mRNA and protein, the significance thresholds were relaxed to FDR < 0.1 and SigB < 0.1 to filter the data. For polysome sequencing, the significant threshold was maintained at $p < 0.1$. Pathway analysis was conducted using gProfileR using the parameters described above.

**Data visualization.** Pathway visualization was done in Cytoscape (v3.6.0). Data visualization was done using Boutroslab.plotting.general (v5.9.2)[127] and ggplot2 (v3.1.0)[128]. Data for the ribbon plot for the network diagram was extracted from the Reactome Functional Interaction Database[129]. Data was visualized using the R package circlize (v0.4.5).

**Reporting summary**
Further information on research design is available in the Nature Portfolio Reporting Summary linked to this article.

## Data availability
All raw and processed data has been deposited into NCBI's Gene Expression Omnibus and are accessible through GEO Series accession numbers: eCLIP (GSE126263), RNA-seq (GSE126337) and polysome profiling-seq (GSE134597). For mass spectrometric proteomic experiments, raw data have been deposited in the ProteomeXchange Consortium via Proteomics Identification (PRIDE), the accession number is PXD012432. Source data are provided with this paper.

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

## Acknowledgements

We are indebted to children and families of the pediatric MB community that serve as the sole motivation of this work. We would also like to thank all the members of the SCC-RI community (Ava Keyvani Chahi, Dr. Blessing Bassey-Achibong, Dr. Maleeha Qazi, Dillon McKenna, Neil Savage, and Dr. Fred Lam) who helped with manpower or to either trouble shoot experimental protocols and critically review the manuscript, and Dr. Eric Van Nostrand for his invaluable advice with eCLIP experiments. We thank the members of the McMaster Illumina Facility for performing the sequencing to a very high standard, the MRI imaging facility, (especially Dr. Kim Desmond) for their expertise and care of mice during the imaging process, and the members of the Stem Cell Unit for their daily care of our mice. M.M.K.S. is supported by the Marta and Owen Boris Foundation, the Ontario Ministry of Health, Ontario Graduate Scholarship (OGS), the McMaster Department of Surgery, and Division of Neurosurgery. J.R. was supported by the Ontario Institute for Cancer Research (OICR) Investigator Award, provided by the Government of Ontario, Natural Sciences and Engineering Research Council of Canada (NSERC) Discovery Grant, and the OICR Brain Tumor Translational Research Initiative. R.W.-R. is supported by funding from the National Cancer Institute (CA159859-07 and P30 CA30199), Alex's Lemonade Stand Foundation, William's Superhero Fund and the McDowell Charity Trust. K.P. is funded by project CZ-OPENSCREEN: National Infrastructure for Chemical Biology (LM2018130) and Bader Phillanthropies. G.W.Y. is Co-Director of the Bioinformatics and Systems biology Graduate Program, Associate Director of the Genetics T32 Training, Grant Deputy Chair of the Sanford Consortium for Regenerative Medicine, Steering group member of Return-to-learn, Co-Director of EXCITE lab, Scientific Director of the SCRM Genomics Core Chair of SCRM Space Committee and Founding member of the UCSD Institute for Genomic Medicine. S.K.S. is supported by the Canadian Institutes of Health Research (CIHR) Operating Grant, Neurosurgical Research and Education Foundation (NREF) and American Association of Neurological Surgeons (AANS), Pediatric Section, the Ontario Institute for Cancer Research (OICR), and McMaster University Department of Surgery. This work was also supported by R35 CA197699 to T.R. R.F. was a recipient of a California Institute for Regenerative Medicine interdisciplinary stem cell training program fellowship and received support from T32 HL086344 and T32 CA009523.

## Author contributions

M.K.S., A.X., C.V., Y.S., C.C., K.H., Y.L., M.R., T.R., R.J.W.-R., G.W.Y., and S.K.S. conceptualized the experiments. M.K.S., A.X., S.X., Y.S., R.F., F.T., D.B., A.A.A., C.C., M.S., P.M., W.G., and I.B. performed the experiments and acquired data. M.K.S., H.Z., E.-C.L., B.Y., D.P., K.B., A.X., K.H., J.M., J.P., Y.L., C.V., K.P., J.R., T.R., G.W.Y., R.J.W.-R., and S.K.S. analyzed and interpreted the data. T.R. and R.F. developed and characterized the Msi1f/f mouse model. M.K.S. wrote the manuscript with significant revisions contributed by J.R., H.Z., C.C., C.V., R.J.W.-R., G.W.Y., and S.K.S. A.F. and S.K.S. provided guidance related to pediatric neuro-oncology pre-clinical trials. R.J.W.-R., G.W.Y., J.R., Y.L., and S.K.S. supervised the study. All authors reviewed the results and commented on the manuscript.

## Competing interests

S.K.S. is a scientific advisor for Century Therapeutics Inc. Her role in the company has been reviewed and is supported by McMaster University. G.W.Y. is co-founder and member of the Board of Directors, on the SAB, equity holder, and paid consultant for Eclipse BioInnovations. T.R. is a founder, member of the Board of Directors and holds executive roles at Tiger Hill Therapeutics. The authors declare no other competing interests.

## Additional information

[1]Centre for Discovery in Cancer Research (CDCR), McMaster University, Hamilton, ON, Canada. [2]Department of Biochemistry and Biomedical Sciences, McMaster University, Hamilton, ON, Canada. [3]Surgery, Faculty of Health Sciences, McMaster University, Hamilton, ON, Canada. [4]Computational Biology Program, Ontario Institute for Cancer Research, Toronto, Canada. [5]Department of Medical Biophysics, University of Toronto, Toronto, Canada. [6]University Health Network, Toronto, ON, Canada. [7]Vector Institute Toronto, Toronto, ON, Canada. [8]Department of Cellular and Molecular Medicine, University of California at San Diego, La Jolla, CA, USA. [9]Stem Cell Program, University of California San Diego, La Jolla, CA, USA. [10]Sanford Consortium for Regenerative

Medicine, La Jolla, CA, USA. [11]Michael G DeGroote School of Medicine, McMaster University, Hamilton, Canada. [12]Tumor Initiation and Maintenance Program, National Cancer Institute-Designated Cancer Center, Sanford Burnham Prebys Medical Discovery Institute, La Jolla, CA, USA. [13]Departments of Pharmacology and Medicine, University of California San Diego School of Medicine, Sanford Consortium for Regenerative Medicine, La Jolla, CA, USA. [14]Donnelly Centre, Department of Molecular Genetics, University of Toronto, Toronto, Canada. [15]Department of Pediatric Oncology, Hematology, and Clinical Immunology, Medical Faculty, University Hospital Düsseldorf, Düsseldorf, Germany. [16]Department of Chemistry, CZ Openscreen, Faculty of Science, Masaryk University, Kamenice 5, 625 00 Brno, Czech Republic. [17]International Clinical Research Center, St. Anne's University Hospital in Brno, 602 00 Brno, Czech Republic. [18]McMaster University, Departments of Pediatrics, Hematology and Oncology Division, Hamilton, Canada. [19]McMaster University, Departments of Neuro-pathology, Hamilton, Canada. [20]Department of Molecular Genetics, University of Toronto, Toronto, Canada. [21]McMaster University, Department of Pediatrics, Hamilton, Canada. [22]Present address: Herbert Irving Comprehensive Cancer Center, Department of Physiology and Cellular Biophysics, Columbia University Medical Center, New York, NY, USA. [23]These authors contributed equally: Robert J. Wechsler-Reya, Gene W. Yeo, Sheila K. Singh.
✉e-mail: rjw2169@cumc.columbia.edu; geneyeo@ucsd.edu; ssingh@mcmaster.ca

