## [Peer Review File · Nature Communications]

Characterization of an RNA binding protein interactome reveals a context-specific post-transcriptional landscape of MYC-amplified medulloblastomaREVIEWER COMMENTS

Reviewer #1 (Remarks to the Author):

In the manuscript from Michelle et al. find that MSI1 is an essential mediator of MYC-amplified medulloblastoma (G3 MB) and reveal cancer-selective MSI1-bound targets through comparative integrative multi-omic analyses. The authors used a MYC-overexpressing mouse model and patient-derived xenografts to identify the essential role for MSI1 in G3 MB. The authors then identified direct binding targets of MSI1 in normal neural and G3 MB cells. Next, they utilized a battery of genome/proteome approaches including: eCLIP, RNA-seq, polysome profiling and LC-MS on control and shMSI1 G3 MB cells. This allowed them to unbiasedly and systematically characterize the MSI1 post-transcriptional regulon in G3 MB at the transcriptomic, translational and proteomic levels. After comparative integrative multi-omic analyses of these datasets, they reveal three targets including HIPK1 as a potential downstream cancer-selective targets in G3 MB. HIPK1 depletion reduced tumorigenesis and was suggested to be a new potential therapeutic target in MB.

Overall, this extensive and impressive set of data provides a comprehensive examination of the MSI1 regulon and identifies a new therapeutic target in MB. Additionally, considering the other data in the field this manuscript is important to document both the functional role in both MB and normal neural stem cells. Despite the other studies this manuscript provides a significant addition to the literature by providing a set of new and important datasets.

The manuscript can be further improved with following comments:

Major comments:

1. The authors should add discussion about the phenotypes in the MSI1 Knockout mice from their studies and from previous published work. Have they performed their own characterization of the in vivo normal phenotype on NSCs? This would more strongly support their statements on the role for MSI1 in the normal context. They should also cite Chavali et al. Science 2017 where they identify two Turkish families with a MSI1 mutation that blocks RNA binding and demonstrate microcephaly phenotypes in people.
2. The conclusions that "inhibiting MSI1 as therapeutics should be avoided in childhood" should also include the possibility for a path as there could still be a therapeutic index if MB are overexpressing as many chemotherapeutic drugs also hit normal tissue and yet are still effective in treating pediatric cancers.
3. After their multi-omic overlaps they identify three targets that are direct MSI1 targets that are altered on the polysome, RNA-seq and protein. Is the proteomics data missing many of the proteins due to low abundance? Can the authors discuss this surprising finding that most targets are not changed on the translational level or discuss the issues with these approaches?
4. The authors should cite and discuss Katz et al. Elife 2014 as they do provide a multiomic analysis of MSI in neural stem cells and compare their findings.
5. The authors indicate that HIPK1 is not a binding target in normal stem cells is that because it is not expressed? Can they speculate why if not? How important is HIPK1 for MSI1's function? Can the authors validate that HIPK1 is reduced in a western blot after MSI1 modulation? The authors should overexpress HIPK1 in shMSI1 to see if it can partially rescue some of the MSI1 knockdown phenotypes.
6. What's the role of TMEM33 and SFXN3 in G3 MB and NSCs?
7. The authors mention MSI2 is a target and do they see any compensatory upregulation since many of their targets are likely shared and even show overlaps with their previous study in the blood.

Minor comments:

1. Some of the supplemental data could go in the main figures such as MSI1 kd in normal context and HIPK1 experiments. The survival curve in figure 1a could go move to supplementary.
2. There are some mislabeling and typos with the figures and figure legends. Like Fig1f is Pathways analysis of MSI1 eCLIP data, it should be at Fig 2 and should be described in the main text; Fig 2f Volcano plot is missing; Supp Fig 2b figure legends is not matching the figure; Supp

Fig 9b figure legends is not matching the figure. Additionally, Supp Fig 2g; 2i should be cre or Msi-KO instead of shMsi1. Supp Fig8c have been cut off a little on the bottom.

3. Fig 3d, is there a significant difference between control and shMSI1?
4. The author should describe what mCherry is in Supp Fig 2d at the figure legend.
5. For HIPK1 CRISPR knockout they should also level of deletion in the normal context in Supplementary Figure 11.

Reviewer #2 (Remarks to the Author):

In the manuscript entitled "Characterization of an RNA binding protein interactome reveals the targetable post-transcriptional landscape of MYC-amplified medulloblastoma" the authors identified Musashi-1 (MSI1) RNA targets in MB cell lines using multi-omic MSI1-eCLIP, RNA-seq, polysome profiling and TMT-MS. The molecular characterization of MSI1 functions and the identification of the putative targets is extensive and well performed. However, the paper lacks experiments/analysis that confirm the relevance of their observations in human patients and in a well-established Group3 Medulloblastoma mouse model.

Major points:

- 1) Since the manuscript mainly focus on finding new MSI1 targets, it is important to test whether the identified MSI1 putative targets are expressed in G3 MB patients and their correlation with MSI1 expression/activity. In my opinion, this analysis will validate the importance of the manuscript findings and it will increase the clinical impact of the results.
- 2) The manuscript is based on the use of a brain cancer model generated by overexpression of Myc and dominant negative p53 (MP) in mouse cerebellar progenitors. However, p53 is rarely mutated in human primary Group3 MB, therefore the authors should confirm their data in a well-established mouse model of Group3 MB (i.e. Northcott et al., 2014, Vo et al., 2017, Ballabio et al., 2020). It would be crucial to determine whether Msi1 is necessary for tumor formation/propagation (i.e. Myc+Gfi1 or Myc+Otx2 overexpression), using Msi1flox/flox mice.
- 3) Mouse survivals curves in figure 1E and supplementary 3G are different, probably due to intrinsic differences of the two cell lines. It would be important to characterize the function/expression of MSI1 and its downstreams in the two lines, in trying to understand the bases of these differences.
- 4) In line 282 the authors talk about the presumed cell of origin of G3 MB: it would be important to evaluate the endogenous levels of MSI1 during cerebellum development in different cerebellar progenitors that have been proposed as G3 MB cell of origin (i.e. doi: 10.1038/s41467-020-17357-4, doi: 10.1126/sciadv.abd2781, doi: 10.1038/onc.2017.110).
- 5) Since MS1 is a well-known modulator of Numb/Notch signaling and that Notch signaling has been demonstrated to be involved during G3 MB development (i.e. <https://doi.org/10.1038/s41467-018-06564-9>, doi:10.1126/sciadv.abd2781) the authors need to show if Notch activation can rescue the loss of MSI1.
- 6) In the title "Characterization of an RNA binding protein interactome reveals the targetable post-transcriptional landscape of MYC-amplified medulloblastoma" the word "targetable" is not appropriate. In fact, the drugs tested in the cell lines need to be tested in vivo to understand their possible relevance to treat G3 MB. In Figure 1B and Supplementary Figure 11F the authors showed striking results with complete rescue of the cancer formation upon MSI1 and HIPK1 loss of function. These results show a very strong impact on mice survival, and it would be important for the brain cancer field to validate their relevance by using drugs (if available) targeting at least HIPKs (i.e. Nature Communications, <https://doi.org/10.1038/s41467-021-26935-z>) in a relevant mouse model (see point 2).

Minor points:

- 1) Supplementary Figure 8c, editing issue.

Reviewer #3 (Remarks to the Author):

The manuscript by Kameda-Smith et. al. describes a comprehensive study, indicating MSI1 plays an essential role in both G3 MB cancer stem cells and normal neural stem cells, directly targeting MSI1 which may cause severe side effects since childhood brain tumor patients are during critical neural development. Instead of targeting MSI1, the alternative method is to target cancer specific MSI1 targets, which might spare neuron stem cell function and abrogate G3 MB cells. The authors integrated multi-omics data from different platforms including MSI1-eClip, RNA-seq, polysome profiling, and proteomics using a G3 MB cell line (SU-MB002) and an embryonic cerebellar NSCs (NSC201cb) and suggested HIPK1 is the most promising therapeutic target of MSI1. This is an interesting topic; however, the statistical evidence and experiment validation to support the conclusions remain relatively limited.

Major comments:

1. Fig2F,3C, and 3G only show subtle differences regarding folder change. How were the cutoffs defined to select the "significant" genes? Since so many different platforms were used in this manuscript, suggest the authors describe the information in a table. The ranking heatmap(fig4a) shows the ranking number varied a lot among different profiling platforms. The authors should add more details (i.e. what value is used to generate the ranked gene list, how to deal with missing values) to the method section about how to apply Robust rank aggregation (RRA) to integrate multi-omic studies.
2. The rationale and evidence to support HIPK1 is the downstream target of MSI1 and the promising therapeutic target in G3 MB is weak. HIPK1 expression was positively correlated with MSI1 inhibition, but its protein level was negatively regulated by MSI1 knock-down in the SU-MB002 cell line. However, supFigure9b shows a strong positive correlation between MSI1 and HIPK1 expression. The authors could comment on the inconsistency between the results and if the analysis results from the SU-MB002 cell line could be applied to other G3 MB models or G3 patients. CRISPR knock-out HIPK1 in SU-MB002 and HDMB03 significantly reduced proliferation and secondary sphere, however, only MB002 HIPK1 KO was tested in vivo with limited mouse number in each condition. I was wondering if the experiment in supFig11f was repeated using MB002 or if any similar experiment was tested in any other MB G3 models such as HDMB03 or other well-established cell lines.

Minor comments:

1. Figure1a, double-check if the x-axis unit (day) is correct. How is the high and low value defined? If comparing the top 25% and low 25% of patients, is there any difference?
2. Figure3h, the x-axis shows maximum $-\log_2 P\text{value}=4$, which means the all the p values > 0.05 ($-\log_2(0.05) = 4.32$)?
3. Figure 4b, eCLIP section: what is the meaning of the y-axis. Why do the three genes have similar logFC and $-\log_{10}(\text{IDR})$ but the plots look so different.
4. Supplementary Figure9 b legend did not match with the plot.

Reviewer #1

In the manuscript from Michelle et al. find that MSI1 is an essential mediator of MYC-amplified medulloblastoma (G3 MB) and reveal cancer-selective MSI1-bound targets through comparative integrative multi-omic analyses. The authors used a MYC-overexpressing mouse model and patient-derived xenografts to identify the essential role for MSI1 in G3 MB. The authors then identified direct binding targets of MSI1 in normal neural and G3 MB cells. Next, they utilized a battery of genome/proteome approaches including: eCLIP, RNA-seq, polysome profiling and LC-MS on control and shMSI1 G3 MB cells. This allowed them to unbiasedly and systematically characterize the MSI1 post-transcriptional regulon in G3 MB at the transcriptomic, translational and proteomic levels. After comparative integrative multi-omic analyses of these datasets, they reveal three targets including HIPK1 as a potential downstream cancer-selective target in G3 MB. HIPK1 depletion reduced tumorigenesis and was suggested to be a new potential therapeutic target in MB.

Overall, this extensive and impressive set of data provides a comprehensive examination of the MSI1 regulon and identifies a new therapeutic target in MB. Additionally, considering the other data in the field this manuscript is important to document both the functional role in both MB and normal neural stem cells. Despite the other studies this manuscript provides a significant addition to the literature by providing a set of new and important datasets.

The manuscript can be further improved with following comments:

Major comments:

1. The authors should add discussion about the phenotypes in the MSI1 Knockout mice from their studies and from previous published work. Have they performed their own characterization of the *in vivo* normal phenotype on NSCs? This would more strongly support their statements on the role for MSI1 in the normal context. They should also cite Chavali *et al.* Science 2017 where they identify two Turkish families with a MSI1 mutation that blocks RNA binding and demonstrate microcephaly phenotypes in people.

We thank the reviewer for their comment, unfortunately there are no *in vivo* normal phenotype characterizations on mouse NSC. However multiple mouse studies have suggested the vital role in normal neural development as described in the text previously. We have now added to our discussion a description of the MSI1 KO phenotypes and additional references as suggested on page 22 and a discussion of Katz *et al*'s findings of MSI1's role in their Dox-inducible Msi1 or Msi2 allele in cortical NSCs from a conditional Tamoxifen-inducible Cre KO mouse. Katz *et al* found in NSCs overexpression of Msi1 yielded a handful of genes with very large changes in ribosome occupancy. Specifically, they noted Msi1 represses translation of the Notch ligand Jagged1 and regulates translation of RBPs by triggering changes in pre-RNA splicing. Interestingly, Katz *et al* too found no change in the previously reported Msi target Numb after Msi1 over-expression but a significant change in the function of a Notch ligand, Jag1.

2. The conclusions that “inhibiting MSI1 as therapeutics should be avoided in childhood” should also include the possibility for a path as there could still be a therapeutic index if MB are overexpressing as many chemotherapeutic drugs also hit normal tissue and yet are still effective in treating pediatric cancers.

We agree with the reviewer's comment and have revised the statement to “While this observation contrasts existing literature⁵⁷, our results suggest that MSI1 may play an essential role in the developing human nervous system and, for this reason, it should be weighed if considered as a direct therapeutic target for treatment of childhood MB” on page 7.

Response to Reviewers

3. After their multi-omic overlaps they identify three targets that are direct MSI1 targets that are altered on the polysome, RNA-seq and protein. Is the proteomics data missing many of the proteins due to low abundance? Can the authors discuss this surprising finding that most targets are not changed on the translational level or discuss the issues with these approaches?

We thank the reviewer for the comment. Our analysis identified 8547 detected protein IDs of which 6150 had a gene ID that we could use (anecdotally is typical of good quality proteomics) with no missing data. Then there were 856 significantly different hits using the Significance B threshold - which is a quite a few of hits considering we compared 2 samples to 2 samples. In our experience, proteomics datasets tend to have smaller effect sizes and thus result in very few hits.

The surprising finding that most targets are not changed at the translational level may be secondary to the limitations of polysome profiling and sequencing which identifies transcripts associated with the polysome fraction, but this does not mean biologically the transcript will be translated and may also be translationally stalled as a response to RNA binding proteins. This data can be inferred from the differences in the changes after shMSI1 KD in SU_MB002 in the polysome profiling and sequencing platform and the mass spectrometry experiments to quantify changes at the proteomic *milieu*. Our RRA analysis takes these factors into account to identify top candidates for further interrogation.

4. The authors should cite and discuss Katz *et al.* Elife 2014 as they do provide a multiomic analysis of MSI in neural stem cells and compare their findings.

Thank you for directing us to Katz et al's work. As commented in point # 1, we have added this reference to our manuscript to describe the Msi1 KO NSC phenotype on page 21.

5. The authors indicate that HIPK1 is not a binding target in normal stem cells is that because it is not expressed? Can they speculate why if not? How important is HIPK1 for MSI1's function? Can the authors validate that HIPK1 is reduced in a western blot after MSI1 modulation? The authors should overexpress HIPK1 in shMSI1 to see if it can partially rescue some of the MSI1 knockdown phenotypes.

Thank you for this comment. HIPK1 was not identified as a binding target in NSC though it is expressed in the cell as evident in Toledo et al's data including 2 NSC cell lines (Suppl Fig 12h) To show the functional relationship between HIPK1 and MSI1 however, we have completed additional overexpression experiments to show that after overexpression of *HIPK1* in G3 MB cell transduced with an shMSI1 construct, a near full rescue of the MSI1 phenotype was observed (Suppl Fig 10g).

6. What's the role of TMEM33 and SFXN3 in G3 MB and NSCs?

Thank you for this comment. The top group of genes from our RRA analysis includes TMEM33, SFXN3 and HIPK1. We have added some details about the role of TMEM and SFXN3 in our discussion.

The role of TMEM33 in G3 MB nor NSC has not been previous studied. In the data available in public repositories however, it is a transmembrane protein that is a known regulator of tubular endoplasmic reticulum network modulating intracellular calcium homeostasis. In the Cavalli data:

Response to Reviewers

In our SU_MB002 CRISPR KO screen using the Toronto KO library (version 3) of gRNA-Cas9 constructs targeting approximately 18,000 protein-coding genes (using BF score is >5 and FDR < 0.05 deemed essential for cell survival), the BF for TMEM33 was 3.5 with FDR 0.0889 (data not shown) and therefore not identified using this approach to be an essential gene in SU_MB002.

The role of SFXN3 has not been studied in either G3 MB nor NSC however, among this protein coding gene's related pathways are as a target of c-myc transcriptional repression. In the Cavalli data:

In our SU_MB002 CRISPR KO screen, the BF for SFXN3 was -11.4 with FDR 0.4799 (data not shown)

Whereas SU_MB002 CRISPR KO screen, the BF for HIPK1 was -5.85 (FDR 0.3984) and MSI1 -6.23 (FDR 0.3217) – interesting to see MSI1 not essential for survival but TMEM is. These results highlight the limitation of large-scale screens in the absence of gene/protein interrogation of changes to the cell with a single gene manipulation. It is clear MSI1 has multiple highly coordinated effects that are vital to the cell. As HIPK1 had a clinical rationale (association with Rubinstein-Taybi Syndrome), we pursued this protein for further drug discovery in both NSC and SU_MB002.

7. The authors mention MSI2 is a target and do they see any compensatory upregulation since many of their targets are likely shared and even show overlaps with their previous study in the blood.

Thank you for your comment regarding MSI2. The Musashi family are indeed comprised of two RNA-binding protein coding genes MSI1 and its paralogue, MSI2. Whilst similar in their genetic code, studies have suggested some divergence of their function in neural tissue including NSC and neural cancers and therefore the ultimate goal of this manuscript to identify the role of MSI1. Interestingly, *in silico* while we found a high transcript abundance of *MSI2* with low *MSI1* levels in the Cavalli *et al* dataset of pediatric MB, the Archer *et al* proteomic dataset did not show an upregulation of MSI2 protein expression with low MSI1 protein expression of MSI1 in pediatric MB again highlighting the importance of analysing the post-transcriptional landscape of targets of RBPs.

Response to Reviewers

In our dataset, we interestingly found MSI1 was uniquely bound to SU_MB002 and not bound to MSI2 transcript in NSC with no significant difference between MSI1 KD and control for fold change (0.006147554) in RNA in SU_MB002 ($p=0.9992681$). This observation is in contrary with Okano's group's conclusion based on double knock-out studies of MSI1 and MSI2 in mouse neural stems that asserted that there is a mutually cooperative role of MSI1 and 2. The authors do state from their observations that while these two paralogues display nearly 90% homology and very similar binding specificity, their roles do seem to differ (Sakakibara, 1996, Sakakibara 2001, Sakakibara 2002).

Minor comments:

1. Some of the supplemental data could go in the main figures such as MSI1 kd in normal context and HIPK1 experiments. The survival curve in figure 1a could go move to supplementary.

We thank the reviewer for this comment. We have since completed the analysis looking at the survival of the MSI1 transcriptomic data as it is associated with survival and found that when comparing the top and bottom quartiles, there is a significant difference and therefore moved Fig 1a to Suppl fig 1a.

2. There are some mislabeling and typos with the figures and figure legends. Like Fig 1f is Pathways analysis of MSI1 eCLIP data, it should be at Fig 2 and should be described in the main text; Fig 2f Volcano plot is missing; Supp Fig 2b figure legends is not matching the figure; Supp Fig 9b figure legends is not matching the figure. Additionally, Supp Fig 2g; 2i should be cre or Msi-KO instead of shMsi1. Supp Fig 8c have been cut off a little on the bottom.

Thank you for your attention to detail and apologize for the errors. Fig 1f's eCLIP pathways analysis is described on page 11 where the reference to the figure has been changed from Fig 2f to 1f). The volcano plot of Fig 2f unfortunately did not export properly when uploaded onto the online system and we have ensured that this is now visible. Supplementary fig 2b has been amended on p46 and ensured the text is also correct in the main manuscript. Supp Fig 2i has been amended to CRE instead of shMSI1. Suppl Fig 8c may also have had a similar export figure issue to 2f. We will ensure adequate review prior to resubmission.

3. Fig 3d, is there a significant difference between control and shMSI1?

Fig 3d depicts SuNSET assay after normalization that shows a trend towards increased nascent polypeptide production in sh*MSI1* inhibited samples in SU_MB002 and NSC201. However, the difference is not significant.

4. The author should describe what mCherry is in Supp Fig 2d at the figure legend.

Thank you for this comment. We have added the statement“(mCherry fluorescent protein chromophore used to identify cells with both myc amplification and p53 mutation)” **to the figure legend for Suppl Fig 2.**

5. For HIPK1 CRISPR knockout they should also level of deletion in the normal context in Supplementary Figure 11.

To address the reviewer's helpful suggestion, we evaluated editing efficiency by PCR amplification and Sanger sequencing of gRNA editing sites, followed by Tracking of Indels by DEcomposition (TIDE) analysis of *HIPK1* knockout clones to reference *AAVSI* knockout clones. All *AAVSI* control clones showed no evidence of inserts or deletions at the *HIPK1* gRNA sites when compared to the human *HIPK1* coding sequence. In addition, TIDE analysis displayed negligible levels of aberrant sequences in all control *AAVSI* samples. In contrast, all *HIPK1* knockout clones displayed high levels of aberrant sequences around the gRNA editing site with editing efficiencies of 31.8-88.4% (represented as test sequences in new Suppl Fig 11).

Reviewer #2 (Remarks to the Author):

In the manuscript entitled “Characterization of an RNA binding protein interactome reveals the targetable post-transcriptional landscape of MYC-amplified medulloblastoma” the authors identified Musashi-1 (MSI1) RNA targets in MB cell lines using multi-omic MSI1-eCLIP, RNA-seq, polysome profiling and TMT-MS. The molecular characterization of MSI1 functions and the identification of the putative targets is extensive and well performed. However, the paper lacks experiments/analysis that confirm the relevance of their observations in human patients and in a well-established Group3 Medulloblastoma mouse model. Major points:

1) Since the manuscript mainly focus on finding new MSI1 targets, it is important to test whether the identified MSI1 putative targets are expressed in G3 MB patients and their correlation with MSI1 expression/activity. In my opinion, this analysis will validate the importance of the manuscript findings and it will increase the clinical impact of the results.

We thank the reviewer for their comment and as suggested we sought publicly available datasets to identify if the MSI1 putative targets are correlated with MSI1 expression. Unfortunately to date, no other study has studied direct targets using CLIP protocol on G3 MB to identify putative *in vivo* targets of MSI1. Our data with respect to the targets of MSI1 and expression in G3MB is visualized in our main Fig 4a. Looking at the correlation of the targets and MSI1 abundance at the transcript and the protein *milieu*, we have completed a Spearman Rho's analysis using repositied Cavalli *et al* and Archer *et al*'s patient derived transcriptomic and proteomic data respectively and found the following

Response to Reviewers

2) The manuscript is based on the use of a brain cancer model generated by overexpression of Myc and dominant negative p53 (MP) in mouse cerebellar progenitors. However, p53 is rarely mutated in human primary Group3 MB, therefore the authors should confirm their data in a well-established mouse model of Group 3 MB (i.e., Northcott et al., 2014, Vo et al., 2017, Ballabio et al., 2020). It would be crucial to determine whether Msi1 is necessary for tumor formation/propagation (i.e., Myc+Gfi1 or Myc+Otx2 overexpression), using Msi1flox/flox mice.

We thank the reviewer for their comment, while the incidence of TP53 mutations in Group 3 MB at diagnosis is low we acknowledge that loss of p53 plays a role post treatment and at recurrence which makes the mice model representative of the more aggressive treatment refractory patient tumours (Pei et al, 2012). Furthermore, we have validated the findings from the mouse model in our established patient-derived xenograft models of medulloblastoma using multiple cell lines. Using additional mouse models to further validate our findings would be outside the scope of this study.

3) Mouse survival curves in figure 1E and supplementary 3G are different, probably due to intrinsic differences of the two cell lines. It would be important to characterize the function/expression of MSI1 and its downstreams in the two lines, in trying to understand the bases of these differences.

We are thankful for the reviewer's comments, the one major difference between the 2 cell lines is the existence of p53 mutation. As HDBM03 in Suppl Fig 3g grows *in vitro* and *in vivo* very quickly, differences in survival were only noted with significant reduction in the number of cells injected as compared to the cell line SU_MB002 in Fig 1d. In Suppl Fig 1b we also show via Western blot the differences between HDMB03 and SUMB002 in MSI1 protein level.

4) In line 282 the authors talk about the presumed cell of origin of G3 MB: it would be important to evaluate the endogenous levels of MSI1 during cerebellum development in different cerebellar progenitors that have been proposed as G3 MB cell of origin (i.e. doi: 10.1038/s41467-020-17357-4, doi: 10.1126/sciadv.abd2781, doi: 10.1038/onc.2017.110).

We thank the reviewer for this comment and the references to relevant manuscripts. Unfortunately, raw data with respects to the different cerebellar progenitor cells are not available for analysis and therefore out with the scope of this project.

5) Since MSI1 is a well-known modulator of Numb/Notch signalling and that Notch signalling has been demonstrated to be involved during G3 MB development (i.e. <https://doi.org/10.1038/s41467-018-06564-9>, doi:10.1126/sciadv.abd2781) the authors need to show if Notch activation can rescue the loss of MSI1.

We thank the reviewer for their comments however we as other authors who have carried out CLIP experiments studying MSI1 in neurooncological cells lines, Numb was not a direct binding partner but we did observe downstream changes at the protein level of the Notch signalling pathway. Further investigation into these proteins and their relationship with MSI1 however we feel is outside the scope of our manuscript.

6) In the title “Characterization of an RNA binding protein interactome reveals the targetable post-transcriptional landscape of MYC-amplified medulloblastoma” the word “targetable” is not appropriate. In fact, the drugs tested in the cell lines need to be tested in vivo to understand their possible relevance to treat G3 MB. In Figure 1B and Supplementary Figure 11F the authors showed striking results with complete rescue of the cancer formation upon MSI1 and HIPK1 loss of function. These results show a very strong impact on mice survival, and it would be important for the brain cancer field to validate their relevance by using drugs (if available) targeting at least HIPKs (i.e. Nature Communications, <https://doi.org/10.1038/s41467-021-26935-z>) in a relevant mouse model (see point 2).

We have revised the title as the reviewer suggested to ”Characterization of an RNA binding protein interactome reveals a context-specific post-transcriptional landscape of MYC-amplified medulloblastoma”. The HIPK1 inhibitor tested as the one the reviewer references are pan HIPK inhibitors (Dr. Paruch, our collaborator has been working to develop a HIPK1 specific inhibitor with great difficulty) that are not specific to HIPK1. For this reason, due to the challenges in generating a HIPK1 specific inhibitor we are unable to translate our genetic KO findings with small molecule inhibitors.

Minor points:

1) Supplementary Figure 8c, editing issue.

Thank you for this astute finding. Cyclophosphamide from the figure legend has been change to the appropriate ciclosporin.

Response to Reviewers

Reviewer #3 (Remarks to the Author):

The manuscript by Kameda-Smith et. al. describes a comprehensive study, indicating MSI1 plays an essential role in both G3 MB cancer stem cells and normal neural stem cells, directly targeting MSI1 which may cause severe side effects since childhood brain tumor patients are during critical neural development. Instead of targeting MSI1, the alternative method is to target cancer specific MSI1 targets, which might spare neuron stem cell function and abrogate G3 MB cells. The authors integrated multi-omics data from different platforms including MSI1-eClip, RNA-seq, polysome profiling, and proteomics using a G3 MB cell line (SU-MB002) and an embryonic cerebellar NSCs (NSC201cb) and suggested HIPK1 is the most promising therapeutic target of MSI1. This is an interesting topic; however, the statistical evidence and experiment validation to support the conclusions remain relatively limited.

Major comments:

1. Fig2F,3C, and 3G only show subtle differences regarding folder change. How were the cutoffs defined to select the “significant” genes? Since so many different platforms were used in this manuscript, suggest the authors describe the information in a table. The ranking heatmap(fig4a) shows the ranking number varied a lot among different profiling platforms. The authors should add more details (i.e., what value is used to generate the ranked gene list, how to deal with missing values) to the method section about how to apply Robust rank aggregation (RRA) to integrate multi-omic studies.

We thank the reviewer for their insightful enquiry. Please refer to the table below for the RRA thresholds employed.

Method	Figure	RRA Threshold
eCLIP-seq	NA	IDR < 0.1 log ₂ FC >1
RNA-seq	2F	FDR < 0.1
Polysome profiling	3C	P < 0.1
Proteomics	3G	SigB < 0.1

Data integration of eCLIP, mRNA, polysome-seq and protein datasets was conducted using Robust Rank Aggregation (RRA), a probabilistic approach to aggregating rank-based lists, using the number of protein coding genes as the number of ranked elements and otherwise, default parameters. Genes were ranked by statistical significance. A threshold rho score of < 0.1 was used to curate the final list. For eCLIP, the most significant peak was chosen for each gene. The significance threshold was relaxed to include sites up to $-\log_{10}IDR > 1$ and $\log_2FC > 1$. For mRNA and protein, the significance thresholds were relaxed to $FDR < 0.1$ adj p-value < 0.1 and $SigB < 0.1$ to filter the data. For polysome sequencing, the significant threshold was maintained at $p < 0.1$. Genes were ranked by statistical significance. Pathway analysis was conducted using gProfileR using the parameters described above.

2. The rationale and evidence to support HIPK1 is the downstream target of MSI1 and the promising therapeutic target in G3 MB is weak. HIPK1 expression was positively correlated with MSI1 inhibition, but its protein level was negatively regulated by MSI1 knock-down in the SU-MB002 cell line. However, supFigure9b shows a strong positive correlation between MSI1 and HIPK1 expression. The authors could

Response to Reviewers

comment on the inconsistency between the results and if the analysis results from the SU-MB002 cell line could be applied to other G3 MB models or G3 patients.

We thank the reviewer for their observation with respect to the changes in association with MSI1 knockdown at the various multi-omic *mileu*. To address the first part of the question, MSI1 is an RNA binding protein and as such functionally regulates transcripts to their functional protein product. Transcription of mRNA is one process, however, what we show most importantly in this manuscript is the central role of post-transcriptional modification of gene regulation which is known to be important in the neural system. Additionally, it is important to note that RNA-seq also does not capture nascent mRNA production and therefore can also be misleading. The inconsistency from upregulation of transcript after MSI1 knockdown and the changes observed in both the polysome, and protein levels is exactly what we aimed to highlight in Fig 4b (HIPK1). We have added further clarification to this on p17.

CRISPR knock-out HIPK1 in SU-MB002 and HDMB03 significantly reduced proliferation and secondary sphere, however, only MB002 HIPK1 KO was tested in vivo with limited mouse number in each condition. I was wondering if the experiment in supFig11f was repeated using MB002 or if any similar experiment was tested in any other MB G3 models such as HDMB03 or other well-established cell lines.

The *in vivo* experiment for the CRISPR HIPK1 KO was repeated in both SU_MB002 and HDMB03 experiment with additional clonal KOs in both SU_MB002 and HDM03 cell lines to address the reviewer's concerns. The results have been added to Suppl Fig 12f-g and maintain a significant survival benefit with HIPK1 CRIPSR knockout.

Minor comments:

1. Figure 1a, double-check if the x-axis unit (day) is correct. How is the high and low value defined? If comparing the top 25% and low 25% of patients, is there any difference?

Figure 1a is using the human data from the Cavalli dataset in Gliovis (<http://gliovis.bioinfo.cnio.es/>). The unit has been changed to months after double checking. The high and low values are defined in the code for Cavalli's dataset using the median as the cut off and defining high as above the median value and low as below the median value.

Further analysis not available on the platform when the manuscript was written were calculations for optimal cut off.

Response to Reviewers

Using 9.6 months as the cut off for Kaplan-Meier survival analysis, $p=0.0039$. When comparing the top 25% and low 25%, there again was no significant correlation with survival and mRNA expression

Once the top and bottom quartiles were analysed there was a significant association with MSI1 transcript expression from the Cavalli database. We have moved this figure to Suppl Fig 1a as per Reviewer 1's recommendations.

2. Figure 3h, the x-axis shows maximum $-\log_2 P\text{value}=4$, which means the all the p values > 0.05 ($-\log_2(0.05) = 4.32$)?

Thank you for noting this discrepancy. The logp value should read $\log_2 p$ vs $-\log_2 p$ after reviewing the analysis and has been amended in the Figure 3h and text on page 45.

3. Figure 4b, eCLIP section: what is the meaning of the y-axis. Why do the three genes have similar $\log_2 FC$ and $-\log_{10}(IDR)$ but the plots look so different.

Thank you for requesting clarification on this figure. There was a minor bug with eCLIP data because there can be multiple peaks per gene so there was a discrepancy between the fold change and the plot fold change. This has been amended in both the figure and text.

4. Supplementary Figure9 b legend did not match with the plot.

The legend has been amended to remove the HIPK1 label as it was not part of the original plot from R from Suppl Fig 9b.

REVIEWERS' COMMENTS

Reviewer #1 (Remarks to the Author):

The authors have responded to the critiques.

Reviewer #2 (Remarks to the Author):

The authors did not answer the following main points:

Main point 2) The manuscript is based on the use of a brain cancer model generated by overexpression of Myc and dominant negative p53 (MP) in mouse cerebellar progenitors. However, p53 is rarely mutated in human primary Group3 MB, therefore the authors should confirm their data in a well-established mouse model of Group 3 MB (i.e., Northcott et al., 2014, Vo et al., 2017, Ballabio et al., 2020). It would be crucial to determine whether Msi1 is necessary for tumor formation/propagation (i.e., Myc+Gfi1 or Myc+Otx2 overexpression), using Msi1flox/flox mice.

Answer: We thank the reviewer for their comment, while the incidence of TP53 mutations in Group 3 MB at diagnosis is low we acknowledge that loss of p53 plays a role post treatment and at recurrence which makes the mice model representative of the more aggressive treatment refractory patient tumours (Pei et al, 2012). Furthermore, we have validated the findings from the mouse model in our established patient-derived xenograft models of medulloblastoma using multiple cell lines. Using additional mouse models to further validate our findings would be outside the scope of this study.

Comment: It is not clear if the authors are modeling primary Group3 MB development or relapsed Group3 MB:

- In case they are modeling primary Group3 MB development, they must use the models previously suggested.
- If the authors want to model relapsed Group3 MB they should modify the text accordingly, for example: "we initially used a well-established G3 MB mouse model, driven by overexpression of Myc and dominantnegative p53 (hereafter called the MP model)" should be corrected. Furthermore, they should provide evidence from the literature and with further analysis that their mouse model (MP) indeed represents a reliable model of human relapsed Group3 MB.

Main point 5) Since MS1 is a well-known modulator of Numb/Notch signalling and that Notch signalling has been demonstrated to be involved during G3 MB development (i.e. <https://doi.org/10.1038/s41467-018-06564-9>, doi:10.1126/sciadv.abd2781) the authors need to show if Notch activation can rescue the loss of MSI1.

Answer: We thank the reviewer for their comments however we as other authors who have carried out CLIP experiments studying MSI1 in neurooncological cells lines, Numb was not a direct binding partner but we did observe downstream changes at the protein level of the Notch signalling pathway. Further investigation into these proteins and their relationship with MSI1 however we feel is outside the scope of our manuscript.

Comment: Since MS1 is a well-known modulator of Numb/Notch signalling and that Notch signalling has been demonstrated to be involved during G3 MB development and relapsed G3 (i.e. <https://doi.org/10.1038/s41467-018-06564-9>, doi:10.1126/sciadv.abd2781) the authors need to show if Notch activation/repression can rescue the loss of MSI1. Indeed, some of the results presented could be due to Notch pathway modulation as the authors wrote: "In fact, downregulation of the Notch pathway upon MSI1 inhibition did not follow the MSI1-Numb axis but led to downregulation of key genes associated with the Notch pathway and the polysome fraction (Supplementary Table 11)".

Main point 6) In the title "Characterization of an RNA binding protein interactome reveals the targetable posttranscriptional landscape of MYC-amplified medulloblastoma" the word "targetable" is not appropriate. In fact, the drugs tested in the cell lines need to be tested in vivo to understand their possible relevance to treat G3 MB. In Figure 1B and Supplementary Figure 11F the authors showed striking results with complete rescue of the cancer formation upon MSI1 and HIPK1 loss of function. These results show a very strong impact on mice survival, and it would be important for the brain cancer field to validate their relevance by using drugs (if available) targeting at least HIPKs (i.e. Nature Communications, <https://doi.org/10.1038/s41467-021-26935-z>) in a relevant mouse model (see point 2).

Answer: We have revised the title as the reviewer suggested to "Characterization of an RNA binding protein interactome reveals a context-specific post-transcriptional landscape of MYC-amplified medulloblastoma". The HIPK1 inhibitor tested as the one the reviewer references are pan HIPK inhibitors (Dr. Paruch, our collaborator has been working to develop a HIPK1 specific inhibitor with great difficulty) that are not specific to HIPK1. For this reason, due to the challenges in generating a HIPK1 specific inhibitor we are unable to translate our genetic KO findings with small molecule inhibitors.

Comment: In this manuscript, the element of novelty is that HIPK1 could be a new MSI1 downstream and that HIPK1 could be required for in vivo tumorigenesis of human cancer cells injected in mouse models. Therefore, testing also pan HIPK inhibitors could represent a new perspective for patients. However, I would put Figure S12 as a main Figure.

Reviewer #3 (Remarks to the Author):

The authors addressed most of my concerns. A few minor ones were newly found:

1. Sup figure10 legend did not match well with the figure
2. Fig 1c and Sup figure3g the mouse number is described differently between the figure label and main text or legend.
3. Why the MB03 cell line two control groups' survival times are so different if we compare Sup Figure3g and Sup Figure12g?

Response to reviewers

Reviewer #1 (Remarks to the Author):

The authors have responded to the critiques.

Reviewer #2 (Remarks to the Author):

Main point 2) The manuscript is based on the use of a brain cancer model generated by overexpression of Myc and dominant negative p53 (MP) in mouse cerebellar progenitors. However, p53 is rarely mutated in human primary Group3 MB, therefore the authors should confirm their data in a well-established mouse model of Group 3 MB (i.e., Northcott et al., 2014, Vo et al., 2017, Ballabio et al., 2020). It would be crucial to determine whether *Msi1* is necessary for tumor formation/propagation (i.e., Myc+Gfi1 or Myc+Otx2 overexpression), using *Msi1*^{flox/flox} mice.

Answer: We thank the reviewer for their comment, while the incidence of TP53 mutations in Group 3 MB at diagnosis is low we acknowledge that loss of p53 plays a role post treatment and at recurrence which makes the mice model representative of the more aggressive treatment refractory patient tumours (Pei et al, 2012). Furthermore, we have validated the findings from the mouse model in our established patient-derived xenograft models of medulloblastoma using multiple cell lines. Using additional mouse models to further validate our findings would be outside the scope of this study.

Comment: It is not clear if the authors are modeling primary Group3 MB development or relapsed Group3 MB: In case they are modeling primary Group3 MB development, they must use the models previously suggested. If the authors want to model relapsed Group3 MB they should modify the text accordingly, for example: “we initially used a well-established G3 MB mouse model, driven by overexpression of Myc and dominant negative p53 (hereafter called the MP model)” should be corrected. Furthermore, they should provide evidence from the literature and with further analysis that their mouse model (MP) indeed represents a reliable model of human relapsed Group3 MB.

We thank the reviewer for bringing this main point back up for discussion. Our manuscript uses lines that model MYC-driven MB common in the difficult to treat and manage primary or recurrent G3 MB. Accordingly, we employed the Pei *et al*/model of MYC-amplified, TP53 mutated cerebellar stem cell (MP model) as a robust mouse model of a MYC-amplified G3 MB and validated these findings with a treatment resistant G3 MB cell line (SU_MB002) and a metastatic at presentation primary G3 MB (HD-MB03) in our human cell line experiments. The common thread transcending these 3 models is their MYC amplification which has been identified in multiple primary tissue studies to be an independent risk factor for poor prognosis (Title, Abstract line 2 &6, Page 3 line 13, Page 4 line 2, Page 5 line 11) validating the mouse model's reliability in modelling MYC-amplified G3 MB. To further clarify, we have amended the last paragraph on page 8 from, “Building on observations of *Msi1* deletion in a syngeneic mouse G₃ MB model, we sought to determine if similar tumor suppressive effects would be achievable in human models of G₃ MB” to “Building on observations of *Msi1* deletion in a

syngeneic mouse Myc amplified G3 MB model, we sought to determine if similar tumor suppressive effects would be achievable in human models of MYC amplified G3 MB.”

Main point 5) Since MSI1 is a well-known modulator of Numb/Notch signalling and that Notch signalling has been demonstrated to be involved during G3 MB development (i.e. <https://doi.org/10.1038/s41467-018-06564-9>, doi:10.1126/sciadv.abd2781) the authors need to show if Notch activation can rescue the loss of MSI1.

Answer: We thank the reviewer for their comments. However, we as other authors who have carried out CLIP experiments studying MSI1 in neurooncological cells lines, found that Numb was not a direct binding partner. but we did observe downstream changes at the protein level of the Notch signalling pathway. Further investigation into these proteins and their relationship with MSI1 however we feel is outside the scope of our manuscript.

Comment: Since MSI1 is a well-known modulator of Numb/Notch signalling and that Notch signalling has been demonstrated to be involved during G3 MB development and relapsed G3 (i.e. <https://doi.org/10.1038/s41467-018-06564-9>, doi:10.1126/sciadv.abd2781) the authors need to show if Notch activation/repression can rescue the loss of MSI1. Indeed, some of the results presented could be due to Notch pathway modulation as the authors wrote: "In fact, downregulation of the Notch pathway upon MSI1 inhibition did not follow the MSI1-Numb axis but led to downregulation of key genes associated with the Notch pathway and the polysome fraction (Supplementary Table 11)".

We thank the reviewer for their discussion with respect to the MSI1-Numb-Notch pathway. It is a very interesting association indeed and would benefit from further characterization (i.e., whether it is protein-protein interaction or otherwise with a downstream regulator of MSI1). However, as this manuscript focusses on the techniques used to identify direct MSI1 binding partners in G3 MB and their downstream effects to identify a target for drug discovery, the statement we included in the manuscript was to corroborate the findings of 3 other large scale studies of MSI1 (i.e., Uren *et al* 2015, Sakakibara *et al* 2012, Katz *et al* 2014) and their observation that Numb was not a direct binding partner of MSI1 as previously thought. As experimental techniques to determine *in vivo* function of proteins continue to progress, we felt it was important to also note our findings. Rev 2'sThe suggested experiments would be a very interesting future follow up project outside of the scope of this current studies.

Main point 6) In the title “Characterization of an RNA binding protein interactome reveals the targetable posttranscriptional landscape of MYC-amplified medulloblastoma” the word “targetable” is not appropriate. In fact, the drugs tested in the cell lines need to be tested *in vivo* to understand their possible relevance to treat G3 MB. In Figure 1B and Supplementary Figure 11F the authors showed striking results with complete rescue of the cancer formation upon MSI1 and HIPK1 loss of function. These results show a very strong impact on mice survival, and it would be important for the brain cancer field to validate their relevance by using drugs (if available) targeting at least HIPKs (i.e., Nature

Communications, <https://doi.org/10.1038/s41467-021-26935-z>) in a relevant mouse model (see point 2).

Answer: We have revised the title as the reviewer suggested to "Characterization of an RNA binding protein interactome reveals a context-specific post-transcriptional landscape of MYC-amplified medulloblastoma". The HIPK1 inhibitor tested as the one the reviewer references are pan HIPK inhibitors (Dr. Paruch, our collaborator has been working to develop a HIPK1 specific inhibitor with great difficulty) that are not specific to HIPK1. For this reason, due to the challenges in generating a HIPK1 specific inhibitor we are unable to translate our genetic KO findings with small molecule inhibitors.

Comment: In this manuscript, the element of novelty is that HIPK1 could be a new MSI1 downstream and that HIPK1 could be required for *in vivo* tumorigenesis of human cancer cells injected in mouse models. Therefore, testing also pan HIPK inhibitors could represent a new perspective for patients. However, I would put Figure S12 as a main Figure.

We thank the reviewer for their insightful comment. While you are correct in our discovery of HIPK1 as a downstream target of MSI1 could be a novel target for treatment of G3 MB, the novelty of the manuscript remains leveraging concurrent technology and *in vivo* study of a neural RNA binding protein to identify HIPK1 as a potential target. Moving our Suppl Fig 12 to a main figure unfortunately with the constraints of 4 figures of the journal was not feasible and we did not feel that it significantly altered the message to the reader. Therefore, no changes were made to the figure order.

Reviewer #3 (Remarks to the Author):

The authors addressed most of my concerns. A few minor ones were newly found:

1. Sup figure10 legend did not match well with the figure.

We thank the Reviewer for their Thank you for your attention to detail. We have amended the figure legend to better explain the figure.

2. Fig 1c and Sup figure3g the mouse number is described differently between the figure label and main text or legend.

We thank the Reviewer Thank you for their astute observation on this. We have verified the experiments and would like to clarify that Fig 1c refers to a mouse experiment using SU_MB002 cell line with MSI1 KD, and Supp Fig 3g is our validation using another primary G3 MB cell line, HDMB03 with MSI1 KD showing a similar trend in survival after injection. We have added HD-MB03 to ensure clarity of these lines in the figure legend.

3. Why the MB03 cell line two control groups' survival times are so different if we compare Sup Figure3g and Sup Figure12g?

We thank the Reviewer for their attention to detail. The survival differences between control groups in Suppl Fig 3g (shControl) and Suppl Fig 12g (AAVS KO) is because of difference in cell numbers injected. In the former case, we used only 1000 cells (mentioned in the legend already) and in the latter we used 10,000 cells. We have consistently observed that the survival time of mice are dictated by cell numbers injected. We have now added the cell numbers to the legend for Supp Fig 12g.